JCB Journal of Cell Biology


# MICAL2 enhances branched actin network disassembly by oxidizing Arp3B-containing Arp2/3 complexes

Chiara Galloni[1]*, Davide Carra[1]*, Jasmine V.G. Abella[1], Svend Kjær[2], Pavithra Singaravelu[3,4], David J. Barry[5], Naoko Kogata[1], Christophe Guérin[3,4], Laurent Blanchoin[3,4], and Michael Way[1,6]

**The mechanisms regulating the disassembly of branched actin networks formed by the Arp2/3 complex still remain to be fully elucidated. In addition, the impact of Arp3 isoforms on the properties of Arp2/3 are also unexplored. We now demonstrate that Arp3 and Arp3B isocomplexes promote actin assembly equally efficiently but generate branched actin networks with different disassembly rates. Arp3B dissociates significantly faster than Arp3 from the network, and its depletion increases actin stability. This difference is due to the oxidation of Arp3B, but not Arp3, by the methionine monooxygenase MICAL2, which is recruited to the actin network by coronin 1C. Substitution of Arp3B Met293 by threonine, the corresponding residue in Arp3, increases actin network stability. Conversely, replacing Arp3 Thr293 with glutamine to mimic Met oxidation promotes disassembly. The ability of MICAL2 to enhance network disassembly also depends on cortactin. Our observations demonstrate that coronin 1C, cortactin, and MICAL2 act together to promote disassembly of branched actin networks by oxidizing Arp3B-containing Arp2/3 complexes.**

## Introduction

The formation of branched actin networks by the Arp2/3 complex plays an essential role in a wide variety of cellular processes (Goley and Welch, 2006; Mullins et al., 1998; Svitkina and Borisy, 1999; Vinzenz et al., 2012). The correct functioning of these actin networks depends not only on their appropriate spatial and temporal assembly but also on their disassembly. We have a good molecular understanding of how nucleation promotion factors such as Neural Wiskott-Aldrich syndrome protein and SPIN90 activate the Arp2/3 complex to stimulate actin filament assembly (Fäßler et al., 2020; Shaaban et al., 2020; Zimmet et al., 2020). In contrast, we still lack detailed molecular insights into the mechanisms regulating network disassembly, which will entail the interplay between factors stabilizing Arp2/3 at branch points and those promoting complex dissociation. For example, it is thought that coronin 1B antagonizes the ability of cortactin to stabilize Arp2/3 at branch points, leading to complex dissociation from the mother filament and subsequent actin network disassembly (Cai et al., 2008). However, while coronin 1B preferentially localizes to actin branches, its binding rarely promotes filament debranching in vitro (Cai et al., 2008). This suggests there must be additional mechanisms to enhance the debranching efficiency of coronin 1B in cells.

Originally identified in *Acanthamoeba castellani*, the Arp2/3 complex consists of seven proteins (actin-related proteins Arp2 and Arp3 and Arp2/3 complex subunits ARPC1–ARPC5; Goley and Welch, 2006; Machesky et al., 1994). The importance of the Arp2/3 complex is underscored by its conservation in all eukaryotes, with the exception of some algae, protists, and microsporidia (Beltzner and Pollard, 2004; Berriman et al., 2005; Gordon and Sibley, 2005; Mathur, 2005; Muller et al., 2005). Interestingly, in many higher eukaryotes, several of the Arp2/3 complex subunits are encoded by more than one gene (Abella et al., 2016). For example, in mammals, the Arp3, ARPC1, and ARPC5 subunits exist as two isoforms (Arp3/3B, ARPC1A/B, and ARPC5/5L), which in humans are 91%, 67%, and 67% identical, respectively (Balasubramanian et al., 1996; Jay et al., 2000; Millard

[1]Cellular Signalling and Cytoskeletal Function Laboratory, The Francis Crick Institute, London, UK; [2]Structural Biology Science Technology Platform, The Francis Crick Institute, London, UK; [3]CytoMorpho Lab, Interdisciplinary Research Institute of Grenoble, Laboratoire de Physiologie Cellulaire & Végétale, University of Grenoble-Alpes, Commissariat à l'Énergie Atomique et aux Énergies Alternatives, Centre National de la Recherche Scientifique, Institut National de la Recherche Agronomique, Grenoble, France; [4]CytoMorpho Lab, Institut de Recherche Saint Louis, University of Paris, Institut National de la Santé et de la Recherche Médicale, Commissariat à l'Énergie Atomique et aux Énergies Alternatives, Paris, France; [5]Advanced Light Microscopy Facility, The Francis Crick Institute, London, UK; [6]Department of Infectious Disease, Imperial College, London, UK.

*C. Galloni and D. Carra contributed equally to this paper; Correspondence to Michael Way: michael.way@crick.ac.uk; C. Galloni's present address is Signal Transduction and Tumour Microenvironment Group, Leeds Institute of Medical Research at St James's, Wellcome Trust Brenner Building, St. James's University Hospital, Beckett Street, Leeds, UK.

et al., 2003). The presence of these subunit isoforms raises the possibility that the mammalian Arp2/3 complex is actually a family of eight isocomplexes, each with different properties that are specifically suited to particular cellular functions. Consistent with this idea, recent observations have now begun to demonstrate that different Arp2/3 isocomplexes have distinct regulatory or functional properties. For example, mutations that lead to a severe reduction or loss of ARPC1B expression in humans result in immunodeficiency and inflammation due to defects in cytotoxic T lymphocyte maintenance and activity (Brigida et al., 2018; Kahr et al., 2017; Kuijpers et al., 2017; Randzavola et al., 2019; Somech et al., 2017; Volpi et al., 2019). It is possible that these conditions occur because ARPC1A is absent or expressed at low levels in haemopoietic cell lineages (Kahr et al., 2017; Kuijpers et al., 2017; Randzavola et al., 2019). However, these devastating physiological changes occur even though the level of ARPC1A increases substantially in the absence of ARPC1B (Kahr et al., 2017; Randzavola et al., 2019). This clearly demonstrates that ARPC1A- and ARPC1B-containing Arp2/3 complexes are not equivalent. The ARPC5 isoforms have also been shown to have distinct functions in an in vitro system that recapitulates embryonic and neonatal skeletal muscle development (Falcone et al., 2014; Roman et al., 2017). In this system, ARPC5L-containing, but not ARPC5-containing, Arp2/3 complexes are required for the peripheral positioning of nuclei in myofibers (Roman et al., 2017). The mispositioning of nuclei in skeletal muscle is a hallmark of numerous diseases, including centronuclear myopathies, which result in progressive and/or fatal muscle dysfunction (Jungbluth et al., 2008). ARPC5L-containing complexes, together with γ-actin, function to squeeze nuclei to the myofiber periphery by promoting myofibril cross-linking in a desmin-dependent fashion. The in vitro myofiber assembly system also uncovered that ARPC5 and β-actin are involved in the formation of transverse triads, which are essential to initiate muscle contraction (Roman et al., 2017).

We previously used actin-based motility of vaccinia virus as a model to study the impact of ARPC1 and ARPC5 isoforms on Arp2/3-dependent actin dynamics (Abella et al., 2016). Our analysis of vaccinia-infected cells, together with in vitro pyrene actin assembly assays, demonstrated that Arp2/3 complexes with ARPC1B and ARPC5L are significantly better at stimulating actin assembly than those containing ARPC1A and ARPC5. In addition, branched actin networks formed by ARPC1B- or ARPC5L-containing complexes disassemble approximately twofold slower than those generated by complexes with ARPC1A or ARPC5. This difference reflects the ability of cortactin to preferentially stabilize complexes containing ARPC1B and ARPC5L, but not ARPC1A and ARPC5, against coronin-mediated filament disassembly (Abella et al., 2016).

In contrast to ARPC1 and ARPC5 isoforms, the impact of the Arp3B isoform on the activity and function of the Arp2/3 complex remains to be established. Northern blot analysis indicates that Arp3B is principally expressed in brain, skeletal muscle, and pancreas (Jay et al., 2000). Available proteomic data, however, indicate that Arp3B is widely expressed, albeit at significantly lower levels than Arp3 (http://pax-db.org). For example, in HeLa cells, which we use to analyze actin-based motility of vaccinia, Arp3B is 18.5-fold less abundant than Arp3 (Hein et al., 2015). In the current study, we have used vaccinia virus as a model system to investigate the impact of Arp3 isoforms on the properties of branched actin networks nucleated by the Arp2/3 complex. We have uncovered that complexes containing Arp3B induce the assembly of branched actin networks that are less stable than those nucleated by Arp3. This reduced stability depends on the monooxygenase MICAL2, which is recruited to branched actin networks by coronin 1C and the presence of cortactin on actin branches.

## Results

### Arp3 isoforms confer different cellular properties to Arp2/3

Immunofluorescence analysis of HeLa cells stably expressing GFP-tagged ARP3 and ARP3B reveals that both Arp3 isoforms are recruited to actin tails induced by vaccinia virus 8 h after infection (Fig. 1 A). Consistent with this, GFP-Trap pulldowns on HeLa cell extracts demonstrates that GFP-tagged ARP3 and ARP3B incorporate into endogenous Arp2/3 complexes and also interact with both ARPC1 and ARPC5 isoforms (Fig. 1 B). Quantification of actin tail lengths demonstrates that GFP-tagged Arp3B, but not Arp3, induced the formation of shorter actin tails compared with parental HeLa cells (hereafter referred to as "control" for tail length experiments with stable GFP-Arp3 and Arp3B cells; Fig. 1 A). This suggests that Arp3B may not be as effective as Arp3 in promoting actin polymerization. To investigate if this is the case, we examined the impact of RNAi-mediated depletion of Arp3 isoforms on actin-based motility of vaccinia (Fig. S1 A). As described previously (Di Nardo et al., 2005; Steffen et al., 2006), loss of Arp3 results in a significant reduction of all Arp2/3 complex subunits (Fig. 1 C). Concomitant with this, there was a dramatic decrease in the length, but not the number, of vaccinia-induced actin tails (Figs. 1 A and S1 B). Stable expression of GFP-tagged Arp3 or Arp3B rescues the loss of the other Arp2/3 subunits induced by Arp3 siRNA (Fig. S1 C). However, the reduction in actin tail length induced by Arp3 siRNA could be fully rescued by expression of GFP-tagged Arp3, but not Arp3B (Fig. 1 A). In contrast to loss of Arp3, knockdown of Arp3B had no appreciable impact on the level of Arp2/3 complex subunits (Fig. 1 C). Nevertheless, there was an increase in actin tail length (Fig. 1 D). Furthermore, expression of GFP-Arp3B in cells treated with Arp3 siRNA had the opposite effect as actin tails were short (Figs. 1 A and S1 C). Our observations clearly demonstrate that Arp3- and Arp3B-containing complexes are not equivalent in promoting vaccinia-induced actin polymerization, with Arp3B-containing complexes being less effective.

### Arp3B generates less stable actin networks than Arp3-containing complexes

To examine whether the differences in actin tail lengths are due to Arp3B complexes being less efficient than those with Arp3 in promoting actin polymerization, we performed in vitro pyrene actin assembly assays using isoform-specific recombinant human Arp2/3 complexes produced in insect cells (Figs. 2 A and S1 D). We found that Arp2/3 complexes containing Arp3 or Arp3B were equally efficient at stimulating actin polymerization, with ARPC1B/ARPC5L-containing complexes being better

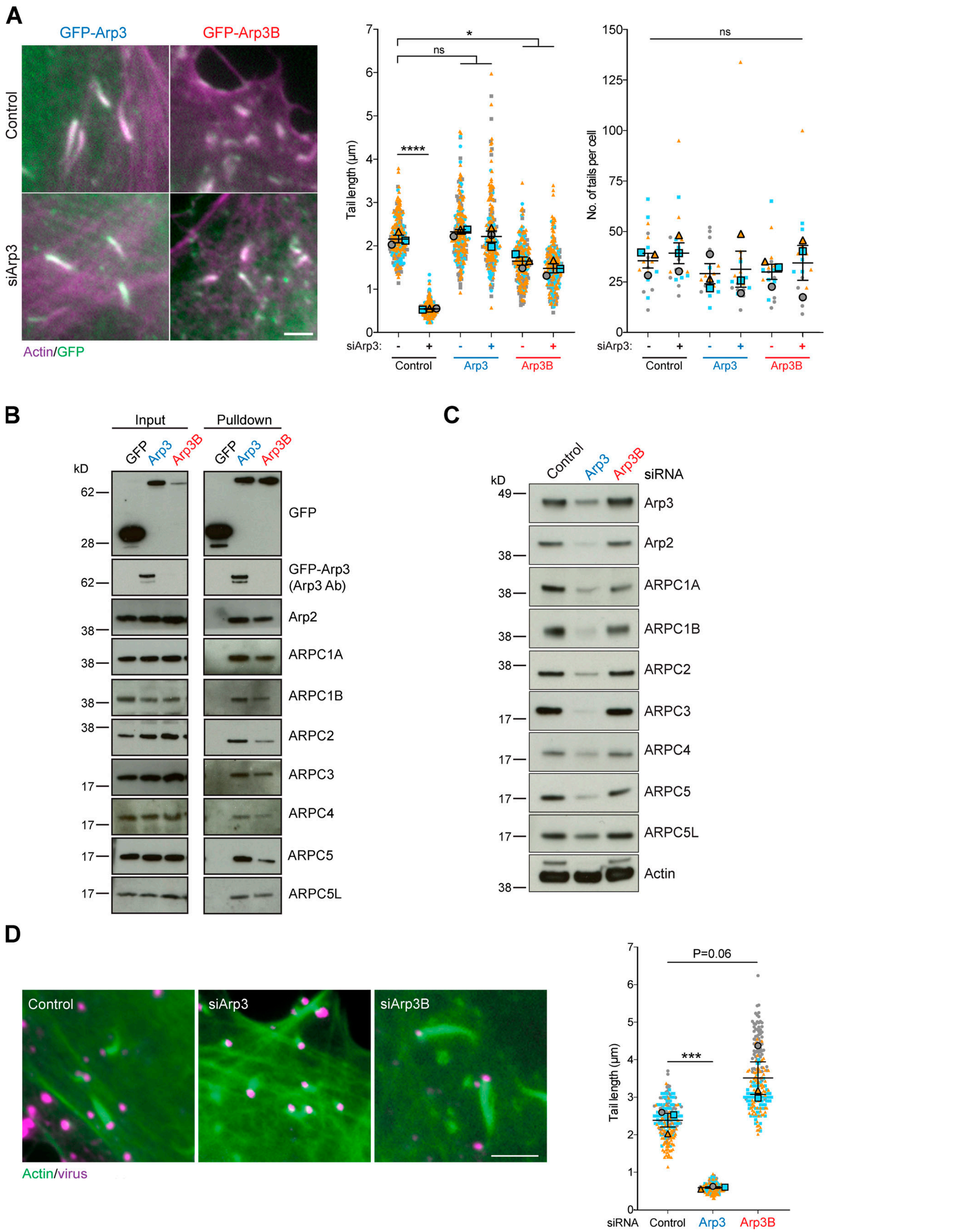


Figure 1. **Arp3 isoforms confer different cellular properties to Arp2/3. (A)** Immunofluorescence images of vaccinia-induced actin tails labeled with Alexa Fluor 568 phalloidin (magenta) together with quantification of their length and number at 8 h after infection in HeLa cells stably expressing RNAi-resistant

GFP-tagged Arp3 (blue) or Arp3B (red) and treated with Arp3 siRNA. **(B)** Immunoblot analysis of GFP-Trap pulldowns from HeLa cells lysates stably expressing GFP, GFP-Arp3, or GFP-Arp3B demonstrates that Arp3 and Arp3B associate with all Arp2/3 complex subunits. The Arp3 antibody does not detect Arp3B. **(C)** Immunoblot analysis of Arp2/3 complex subunits in HeLa cells treated with Arp3 and Arp3B siRNAs (Fig. S1 A shows RT-PCR analysis of the level of Arp3 and Arp3B mRNA relative to control in HeLa cells after knockdown with the individual siRNA from the siGenome pools against Arp3 and Arp3B). **(D)** Immunofluorescence images of virus (magenta) induced actin tails labeled with Alexa Fluor 488 phalloidin (green) together with quantification of their length in cells treated with the indicated siRNAs. All error bars represent SEM from $n$ = 3 independent experiments in which the length of 100 tails or their number in six cells was analyzed per condition. Tukey's multiple comparisons test was used to determine statistical significance; ****, $P < 0.0001$; ***, $P < 0.001$; *, $P < 0.05$. Scale bars = 5 µm.

than those with ARPC1A/ARPC5, as we previously observed (Abella et al., 2016; Fig. 2 A). We also performed total internal reflection fluorescence (TIRF) microscopy to directly visualize Arp2/3-mediated actin assembly and branching (Fig. 2 B and Video 1). Automated quantification with the AnaMorf ImageJ plugin (Barry et al., 2015) reveals there are no appreciable differences in the rate of actin branch assembly by Arp3- or Arp3B-containing complexes (Fig. 2 C).

As the length of an actin tail is determined by the balance between the rate of new actin polymerization and filament loss, we wondered whether the short actin tails induced by Arp3B are due to increased actin disassembly. To investigate this possibility, we took advantage of photoactivatable mCherry-GFP$^{PA}$-actin to measure the rate of actin tail disassembly in cells treated with Arp3B siRNA (Figs. 3 A and S1 E). We found that the loss of Arp3B increases the half-life of actin disassembly from 8.00 ± 0.05 s to 9.80 ± 0.29 s (Fig. 3 B and Video 2 and Video 3). This suggests that Arp3B generates less stable branched actin networks. To examine whether this difference in actin disassembly reflects the turnover rate of Arp3- and Arp3B-containing complexes, we generated HeLa cell lines stably expressing photoactivatable GFP$^{PA}$-Arp3 isoforms (Fig. S1 F). Consistent with the reduced actin disassembly in the absence of Arp3B, we found that the half-life of Arp3B in actin tails (6.64 ± 0.17 s) is also significantly lower than that of Arp3 (8.61 ± 0.23 s; Fig. 3, C and D; and Video 4 and Video 5). Our observations demonstrate that Arp3B-containing Arp2/3 complexes generate less stable actin assemblies because they are removed faster from the network than those with Arp3.

### Met293 of Arp3B is required for its cellular properties

Human Arp3 and Arp3B are 91% identical with amino acid differences spread throughout their length (Fig. 4 A). To narrow down the region and/or amino acids responsible for the difference in actin network stability, we examined the impact of a series of RNAi-resistant Arp3/3B chimeras on actin tail length in HeLa cells treated with Arp3 and Arp3B siRNAs (Fig. 4, B and C). Analysis of infected cells reveals that residues 281–418 of Arp3B are sufficient to confer the short actin tail phenotype (Fig. 4 D). This region of Arp3B contains 10 conservative and 3 nonconservative substitutions between the two proteins (Fig. 4 A). Examination of the structure of Arp2/3-containing Arp3 in its inactive and active state reveals that two nonconserved substitutions (Arp3 Thr293-Pro295 and Arp3B Met293-Ser295) are present on the surface of Arp3 in a flexible loop that is positioned close to Arp2 (Fäßler et al., 2020; Robinson et al., 2001; Shaaban et al., 2020; von Loeffelholz et al., 2020; Zimmet et al., 2020; Figs. 5 A and S2). To investigate their possible role in mediating isoform differences, we exchanged residues 293 and 295 between Arp3 and Arp3B to generate cell lines stably expressing GFP-tagged Arp3$^{MS}$ and Arp3B$^{TP}$ (Fig. 5, B and C). Quantification of actin tail lengths in cells expressing these mutants reveals that GFP-tagged Arp3$^{MS}$ and Arp3B$^{TP}$ switched phenotypes, inducing the formation of short and long actin tails, respectively (Fig. 5, D and E). Vaccinia-induced actin tails remained long when the single T293M or P295S Arp3B substitutions were introduced into Arp3. In contrast, actin tails remained short when Ser295 in Arp3B was changed to proline but became long when Met293 was substituted to threonine (Fig. 5, D and E). These experiments demonstrate that Met293 is essential for Arp3B to induce short actin tails.

### The Arp3B actin tail phenotype depends on MICAL2, but not MICAL1

Oxidation of actin Met44 and Met47 to methionine sulfoxide (Met-SO) by MICAL proteins, which have three different isoforms in humans, promotes actin filament disassembly (Frémont et al., 2017b; Hung et al., 2011). Given this, we wondered whether the Arp3B-dependent actin tail phenotype depends on oxidation of Met293. To investigate this possibility, we replaced Thr293 in Arp3 and Met293 in Arp3B with glutamine to mimic the Met-SO state (Aledo, 2019; Bigelow and Squier, 2011; Fig. S3 A). We found that in both cases, the stable expression of these glutamine mutants resulted in the formation of shorter actin tails (Fig. 6 A). Consistent with the notion that Met293 in Arp3B can be oxidized by MICAL proteins, we observed that GFP-tagged MICAL2, but not MICAL1, is recruited to actin tails (Figs. 6 B and S3 B; and Video 6 and Video 7). It is also striking that GFP-MICAL2 is not recruited immediately behind the virus but further down the actin tail (Fig. 6 B), as we previously observed for coronin (Abella et al., 2016). To investigate whether MICAL2 is required to promote short actin tails by enhancing actin network disassembly, we analyzed the impact of MICAL2 depletion on actin turnover in vaccinia infected mCherry-GFP$^{PA}$-actin stable cells (Fig. S3 C). We found that knockdown of MICAL2 increases the half-life of actin disassembly in tails by ~2 s (Fig. 6 C). In contrast, loss of MICAL1 had no impact on the rate of actin disassembly in the tail, consistent with its lack of recruitment. Moreover, depletion of MICAL2 or Arp3B alone and together resulted in a similar increase in actin tail length (Figs. 6 D and S3 D), suggesting that both proteins are operating in the same pathway. Consistent with this, loss of MICAL2 suppressed the short actin phenotype induced by the overexpression of GFP-Arp3B (Figs. 6 E and S3 E). To confirm that MICAL2 is oxidized we raised an antibody against a peptide corresponding to residues 289–298 of Arp3B containing oxidized Met293.

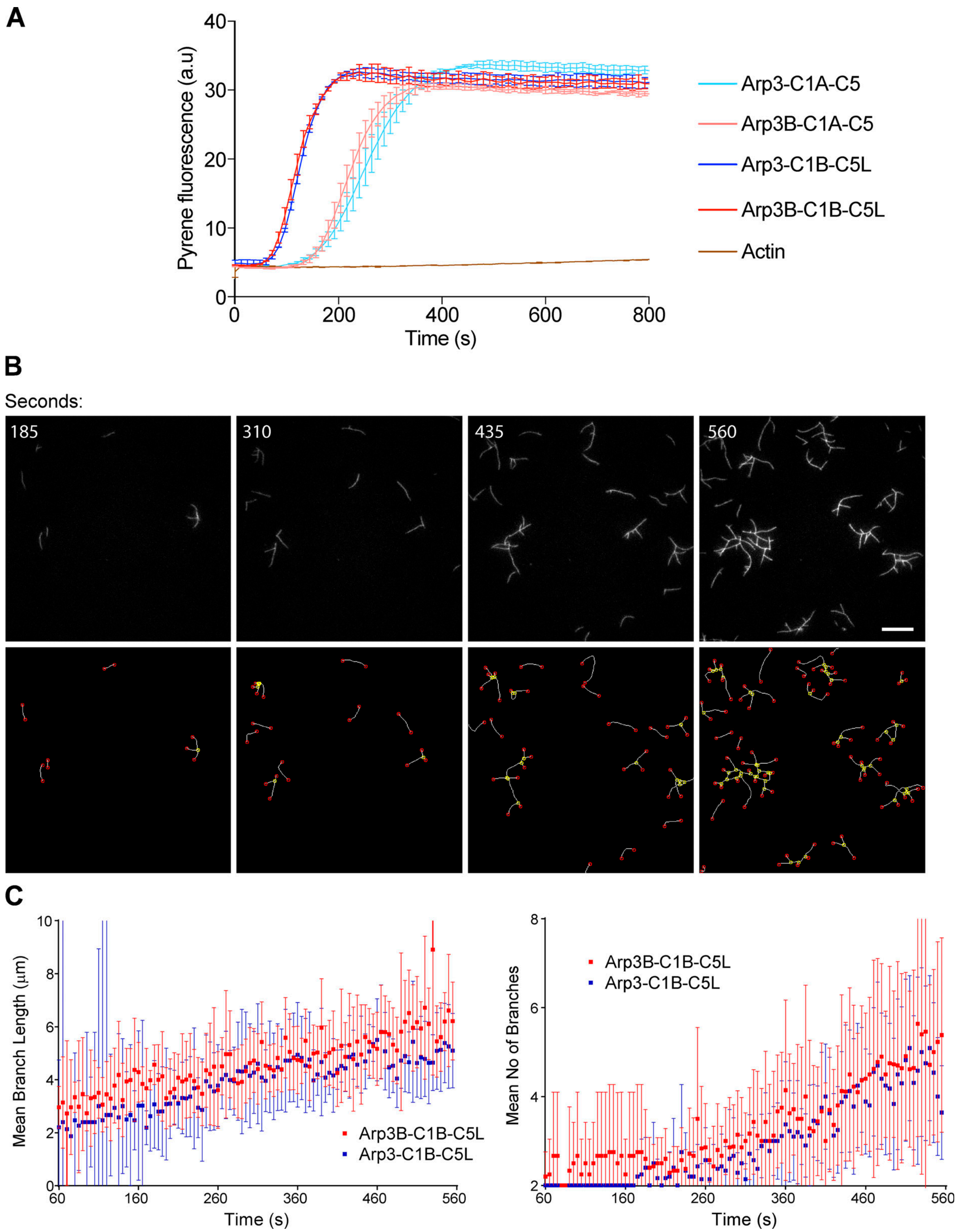


Figure 2. **Arp3- and Arp3B-containing complexes have the same actin nucleating activity. (A)** In vitro pyrene actin assays using 12.5 nM of the indicated recombinant Arp2/3 complexes reveal Arp3 and Arp3B containing complexes nucleate actin equally efficient, regardless of the ARPC1A/ARPC5 and ARPC1B/ARPC5L background. The error bars represent SEM from $n$ = 2 independent experiments. **(B)** The top row shows representative in vitro TIRF microscope images of branched actin formation using 2.5-nM Arp2/3 complexes containing Arp3B, ARPC1A, and ARPC5 at the indicated times (see Video 1). $n$ = 2 independent experiments. Scale bar = 15 µm. The bottom row shows the same panels after automatic detection of filament branches (yellow nodes) and ends (red nodes) with AnaMorf ImageJ plugin. **(C)** Quantification of mean branch number and filament length for the indicated Arp2/3 complexes from one of the two independent TIRF experiments. Each data point represents the average of 14 and 15 actin "structures" for Arp3B-C1B-C5L and Arp3-C1B-C5L, respectively, although there are fewer structures to analyze at the beginning of the experiment than at the end.

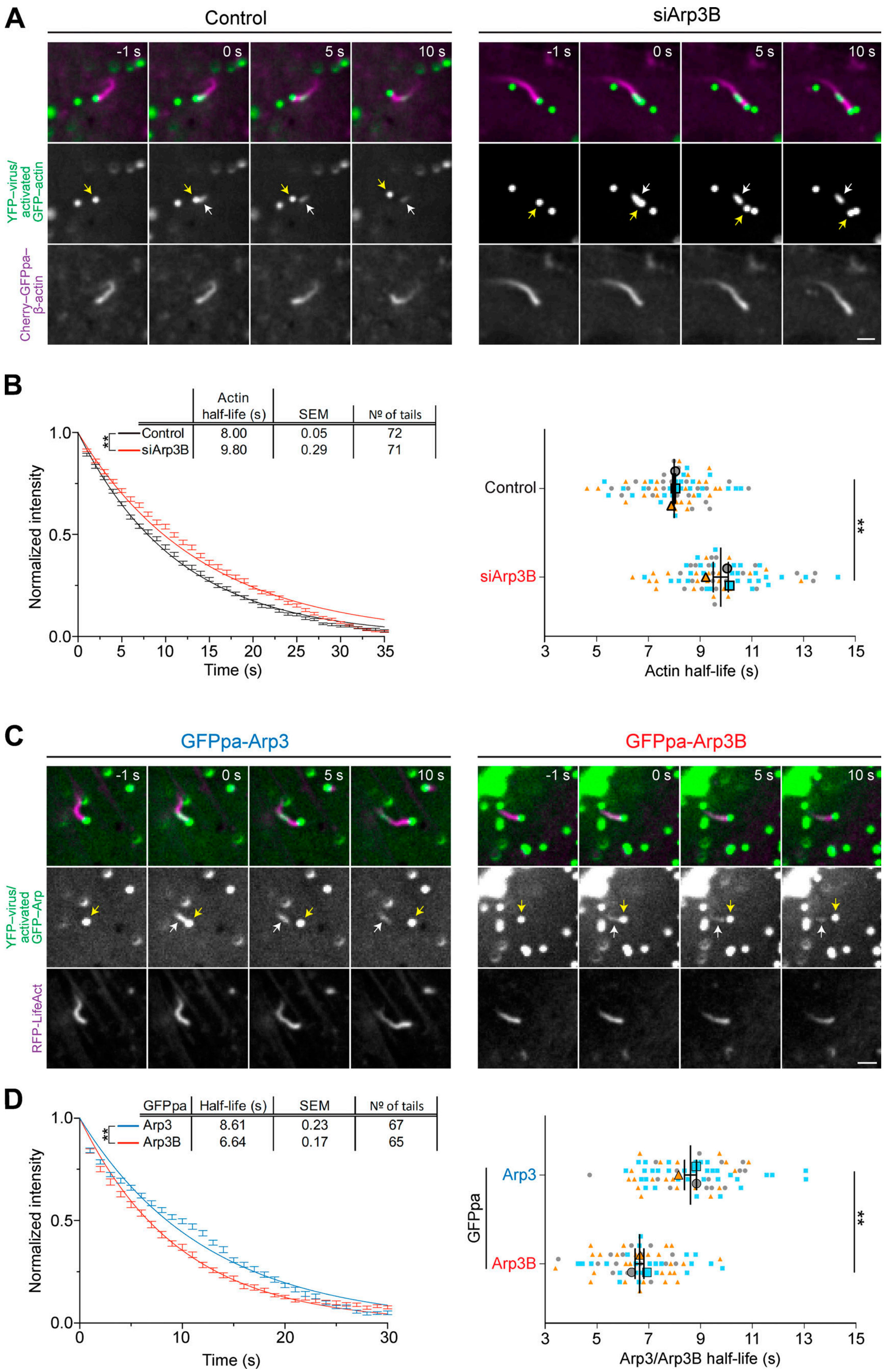


Figure 3. **Arp3B reduces the stability of actin networks. (A)** Image stills from Video 2 and Video 3 at the indicated times in seconds before and after photoactivation showing the movement of YFP-tagged virus (yellow arrow) in HeLa cells stably expressing Cherry-GFP$^{PA}$-β-actin (magenta, before

photoactivation) in control and Arp3B RNAi–treated cells. The white arrow indicates the position of photoactivated Cherry-GFP$^{PA}$-β-actin at time 0 (green). **(B)** Quantification of the half-life of photoactivated GFP$^{PA}$-β-actin in actin tails in Arp3B RNAi–treated cells. These data were collected at same time as the data in Fig. 9 D. **(C)** Image stills from Video 4 and Video 5 at the indicated times in seconds showing the movement of YFP-tagged virus (yellow arrow) in HeLa cells stably expressing LifeAct-RFP (shown in magenta) together with GFP$^{PA}$-tagged Arp3 or Arp3B. White arrows indicate the position of photoactivation of GFP$^{PA}$-tagged Arp3 or Arp3B at time 0 (green). **(D)** Quantification of the half-life of Arp3 and Arp3B in actin tails after activation of the GFP$^{PA}$ tag. For B and D, the left graphs represent the best-fitting curve for each condition (continuous line) together with the average normalized intensity of the GFP signal at every time point (error bars represent the SEM for the indicated number of tails). The right graphs show the mean half-life and SEM from $n$ = 3 independent experiments. Student's $t$ test was used to determine statistical significance; **, P < 0.01. Scale bars = 2.5 µm.

Immunoblot blot analysis of GFP-Trap pulldowns reveals that the Met293-OX antibody detects GFP-tagged Arp3B, but not Arp3 (Fig. 7 A). This signal was lost when Met293 of Arp3B was mutated to Thr293, the equivalent residue in Arp3. The level of Arp3B Met293 oxidation was also reduced when cells were treated with MICAL2 siRNA (Fig. 7 B). Our data demonstrate that MICAL2-mediated oxidation of Arp3B Met293 promotes faster actin network disassembly, resulting in shorter actin tails.

### MICAL2 mediated actin disassembly depends on coronin 1C

Given the similar spatial localization of MICAL2 and coronin 1C in actin tails, we decided to investigate whether there is a connection between the two proteins, especially in light of the role of coronins in promoting disassembly of branched actin networks (Abella et al., 2016; Cai et al., 2008). We found that siRNA-mediated depletion of coronin 1C resulted in a dramatic loss of GFP-MICAL2 recruitment to actin tails (Fig. 8 A and Video 8 and Video 9). In addition, GFP-Trap pulldowns on lysates from cells stably expressing GFP-tagged MICAL1 and MICAL2 revealed that coronin 1C associates with MICAL2, but not MICAL1 (Fig. 8 B). No GFP-MICAL2 is detected in the input, as the protein is expressed at very low level compared with GFP-MICAL1 (Fig. S3 B).

Consistent with our observations that coronin 1C is required for MICAL2 to exerts its effects on branched actin networks, loss of coronin 1C also suppresses the short actin tail phenotype induced by expression of GFP-Arp3B in cells treated with Arp3 and Arp3B siRNA (Fig. 9, A and B). Moreover, depletion of coronin 1C or MICAL2 results in a similar ~3-s increase in Arp3B half-life while having minimal impact on Arp3 dynamics (Fig. 9 C; and Fig. S4, A and B). Concomitant with the increased stability of Arp3B in the actin network, photoactivation experiments with mCherry-GFP$^{PA}$-actin demonstrate that there is a similar increase in the half-life of actin disassembly when Arp3B, coronin 1C or MICAL2 were depleted alone or in a pairwise fashion (Fig. 9 D; and Fig. S4, C and D). Our observations demonstrate that coronin 1C is required to recruit MICAL2 to actin tails to promote the disassembly of Arp3B but not Arp3 actin branches. Enhanced coronin 1C–mediated debranching of oxidized Arp3B containing complexes would account for the faster turnover of Arp3B and the short actin tails generated by its overexpression.

### Cortactin is required for MICAL2 mediated actin disassembly, but not its recruitment

We previously demonstrated that coronin-induced actin tail disassembly depends on the presence of cortactin (Abella et al., 2016). Given this, we examined whether loss of cortactin impacts on the recruitment of MICAL2. We found that both MICAL2 and coronin 1C were still recruited to actin tails when cortactin was depleted (Figs. 10 A and S5 A and Video 10). Nevertheless, loss of cortactin suppresses the short actin tail phenotype induced by expression of GFP-Arp3B but has no impact on Arp3 induced actin tails (Fig. 10 B; and Fig. S5, B and C). This observation predicts that the loss of cortactin will increase the stability of Arp3B in actin tails but have no effect on the half-life of Arp3. Consistent with this hypothesis, photoactivation experiments demonstrate that depletion of cortactin increases the half-life of Arp3B from 6.29 ± 0.33 to 9.57 ± 0.30 s (Fig. 10 C; and Fig. S5, D and E). In contrast, loss of cortactin had no impact on Arp3 dynamics (8.30 ± 0.13 to 8.33 ± 0.35 s when cortactin is depleted). Taken together our observations demonstrate that cortactin in addition to coronin 1C–mediated recruitment of MICAL2 is required to enhance the dissociation of Arp3B-containing Arp2/3 complexes and increase the rate of actin network disassembly.

## Discussion

### Arp3 and Arp3B isocomplexes generate actin networks with different stabilities

We still lack a complete molecular understanding of branched actin network disassembly, including how proteins involved in this process are regulated in a coordinated fashion to dissociate Arp2/3 from the mother actin filament. In addition, it remains to be established whether the Arp2/3 complex also plays a role in coordinating or promoting the disassembly of the actin network in which it is embedded. Our observations examining the actin-based motility of vaccinia virus reveal that MICAL2, coronin 1C, cortactin, and Arp3B-containing Arp2/3 complexes act in a coordinated fashion to enhance disassembly of branched actin networks (Fig. 10 D).

Arp3- and Arp3B-containing Arp2/3 complexes have the same actin nucleating activity regardless of the presence of ARPC1 and ARPC5 isoforms. However, Arp3B-containing Arp2/3 complexes induce the formation of less stable actin networks than those containing Arp3. The loss of Arp3B therefore leads to a significant increase in the persistence of the branched actin network. This effect is not unique to vaccinia-induced actin tails. Previous work identifying regulators of spindle orientation during mitosis noted that loss of Arp3B increases cortical actin thickness by ~35% (Fig. 4, G and H; di Pietro et al., 2017). Although this observation was an aside, it clearly demonstrates that Arp3B-containing Arp2/3 complexes play an important role in promoting turnover of branched actin networks.

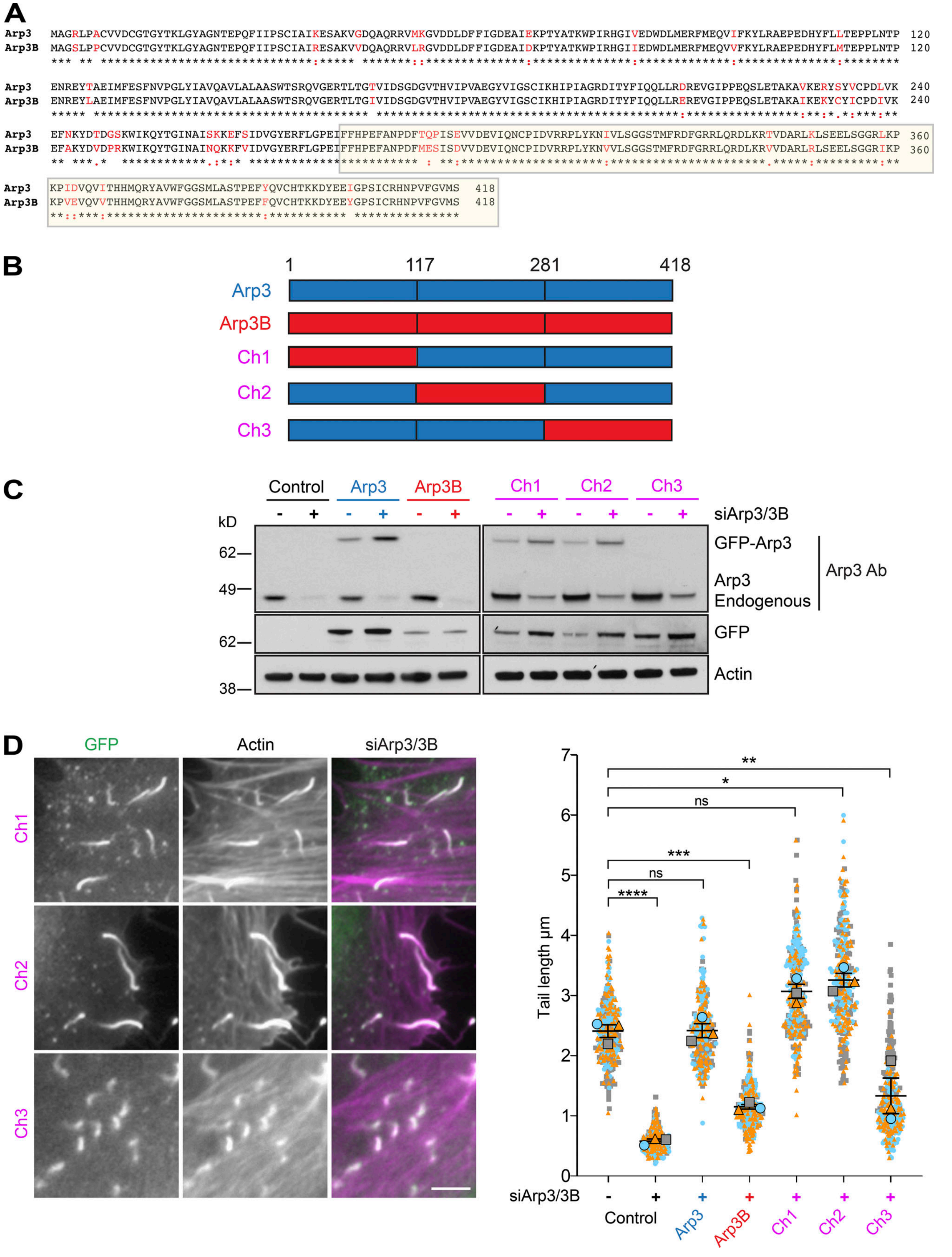


Figure 4. **The C-terminal third of Arp3B promotes the formation of short actin tails. (A)** Alignment of human Arp3 and Arp3B showing conservative residues (black) and nonconservative amino acid differences (red). The C-terminal third of Arp3/3B responsible for long or short actin tail phenotypes is boxed

in yellow. **(B)** Schematic representation of RNAi-resistant human Arp3 (blue) and Arp3B (red) together with the three different chimeras (Ch1, Ch2, and Ch3; blue + red = purple). Residue positions at the splice sites are indicated. **(C)** The immunoblot shows the expression level of RNAi-resistant GFP-tagged Arp3 and Arp3B together with the three different chimeras in HeLa cells treated with Arp3 and Arp3B siRNA. **(D)** Images of actin tails labeled with Alexa Fluor 568 phalloidin (magenta) together with quantification of their length in HeLa cells stably expressing RNAi-resistant GFP-tagged Ch1, Ch2, or Ch3 (green) and treated with Arp3 and Arp3B siRNA. All error bars represent SEM from $n = 3$ independent experiments in which the length of 100 tails was analyzed per condition. Tukey's multiple comparisons test was used to determine statistical significance; **** $P < 0.0001$; *** $P < 0.001$; ** $P < 0.01$; * $P < 0.05$. Scale bar = 5 µm.

### MICAL2 promotes disassembly of actin networks assembled by Arp3B complexes

Humans have three MICAL isoforms (MICAL1–MICAL3), with MICAL2 and MICAL3 being more similar to each other (70% identical) than to MICAL1 (54% and 52% identity, respectively; Frémont et al., 2017b; Hung et al., 2010). All three MICAL isoforms can promote filament disassembly by oxidizing Met44 and Met47 of actin (Grintsevich et al., 2017; Hung et al., 2011; Hung et al., 2013; Lee et al., 2013; Wu et al., 2018). In addition, MICAL-mediated oxidation of actin enhances cofilin-driven filament disassembly (Grintsevich et al., 2017; Grintsevich et al., 2016; Wioland et al., 2021). These observations suggest that the three MICAL isoforms are functionally redundant. However, our observation that the reduced stability of Arp3B-induced actin networks depends on MICAL2, but not MICAL1, clearly indicates that the different MICAL isoforms can have unique cellular functions.

Curiously, full-length MICAL1 does not accelerate the disassembly of actin filaments in vitro, leading to the suggestion that the protein is autoinhibited (Frémont et al., 2017a). This autoinhibition is relieved by the addition of GTP-bound Rab35, which binds to a flat three-helix domain at the MICAL1 C terminus (Frémont et al., 2017a). Consistent with this, all previous in vitro assays examining the interaction of MICAL proteins with actin and/or their impact on filament disassembly have used recombinant proteins lacking extensive portions of the C-terminal half of the molecule (Bai et al., 2020; Grintsevich et al., 2017; Hung et al., 2011; Hung et al., 2013; Hung et al., 2010; Lee et al., 2013; Lundquist et al., 2014; Wioland et al., 2021; Wu et al., 2018). Rab35 is also responsible for recruiting MICAL1 to the abscission site during cytokinesis (Frémont et al., 2017a). Our observation that the recruitment of MICAL2 to vaccinia-induced actin tails depends on coronin 1C is further evidence that MICAL protein localization is highly regulated. It is also possible that coronin 1C both activates and targets MICAL2 to branched actin networks. The inability of MICAL1 to associate with coronin 1C would explain its lack of recruitment and why only MICAL2 promotes disassembly of actin tails. Our results clearly demonstrate that MICAL isoforms are differentially regulated, consistent with the observation that transient expression of HA-tagged MICAL2, but not MICAL1, induces the disassembly of actin stress fibers (Giridharan et al., 2012).

### MICAL2 induces actin network disassembly by oxidizing Arp3B

Our analysis demonstrates that the presence of Arp3B-containing Arp2/3 complexes enhances the ability of MICAL2 to promote actin network disassembly. Furthermore, our photoactivation experiments reveal that the loss of MICAL2, Arp3B, or coronin 1C or any pairwise depletion of these proteins leads to a similar increase in actin network stability. This epistatic analysis suggests that MICAL2 increases network disassembly by primarily promoting debranching of Arp3B-containing Arp2/3 complexes rather than directly oxidizing actin filaments. While unexpected, such a mechanism might be explained by the activation of MICAL2 only at actin branches. Such an activation mechanism appears to involve cortactin, which is required for the short actin tail induced by Arp3B, but is not required for the recruitment of MICAL2 or coronin. Activated MICAL2, bound to actin branches, would only be able to oxidize Arp3B, as Arp3 lacks Met293. This would explain why loss of cortactin reduces the turnover of Arp3B but has no impact on Arp3 dynamics.

Interestingly, substitution of Thr293 in Arp3 with methionine does not result in shorter actin tails. In contrast, substitution of both Thr293 and Pro295 with methionine and serine, respectively, is sufficient to generate short actin tails. There is no consensus motif for MICAL-mediated oxidation of methionine, so we cannot say whether the presence of serine at the +2 position is required for methionine oxidation or if proline in the equivalent position inhibits oxidation of Met293 when its substituted into Arp3. However, substitution of Ser295 in Arp3B with proline does not induce long actin tails (Arp3 phenotype). This suggests that additional residues influence the oxidation of Met293 in Arp3B.

It is maybe not surprising that MICALs can regulate both actin and actin-related proteins given their similar structural fold and high degree of sequence identity. However, the residues targeted by MICAL are different between actin and Arp3B, with the latter having leucine and valine in the equivalent positions of Met44 and Met47 of actin. In contrast, actin contains a MES (Met, Ser, and Glu) motif (residues 269–271) equivalent to the one in Arp3B (residues 293–295), which is also exposed on the surface of the actin filament (Fäßler et al., 2020; Shaaban et al., 2020; von der Ecken et al., 2015). However, there is currently no evidence that Met269 in actin is oxidized (Hung et al., 2011). Arp2 and Arp3 also have LMVG (residues 55–58) and VMKG (residues 51–54) motifs, respectively, that are related to the sequences flanking Met44 in actin (VMVG residues 43–46). Met52 of Arp3 is surface exposed in the "open" inactive Arp2/3 structure but is buried in the cryo-electron tomography structure of activated Arp2/3 at actin branches (Fäßler et al., 2020; Robinson et al., 2001; von Loeffelholz et al., 2020; Zimmet et al., 2020). In contrast, Arp2 Met56 is surface exposed in Arp2/3 bound at branch points, although this structure lacks cortactin (Fäßler et al., 2020). Further analysis is required to determine whether Met56 of Arp2 is oxidized and its potential impact on the function of the Arp2/3 complex. Notwithstanding this, our data clearly demonstrate that MICAL2 only enhances debranching of complexes containing Arp3B.

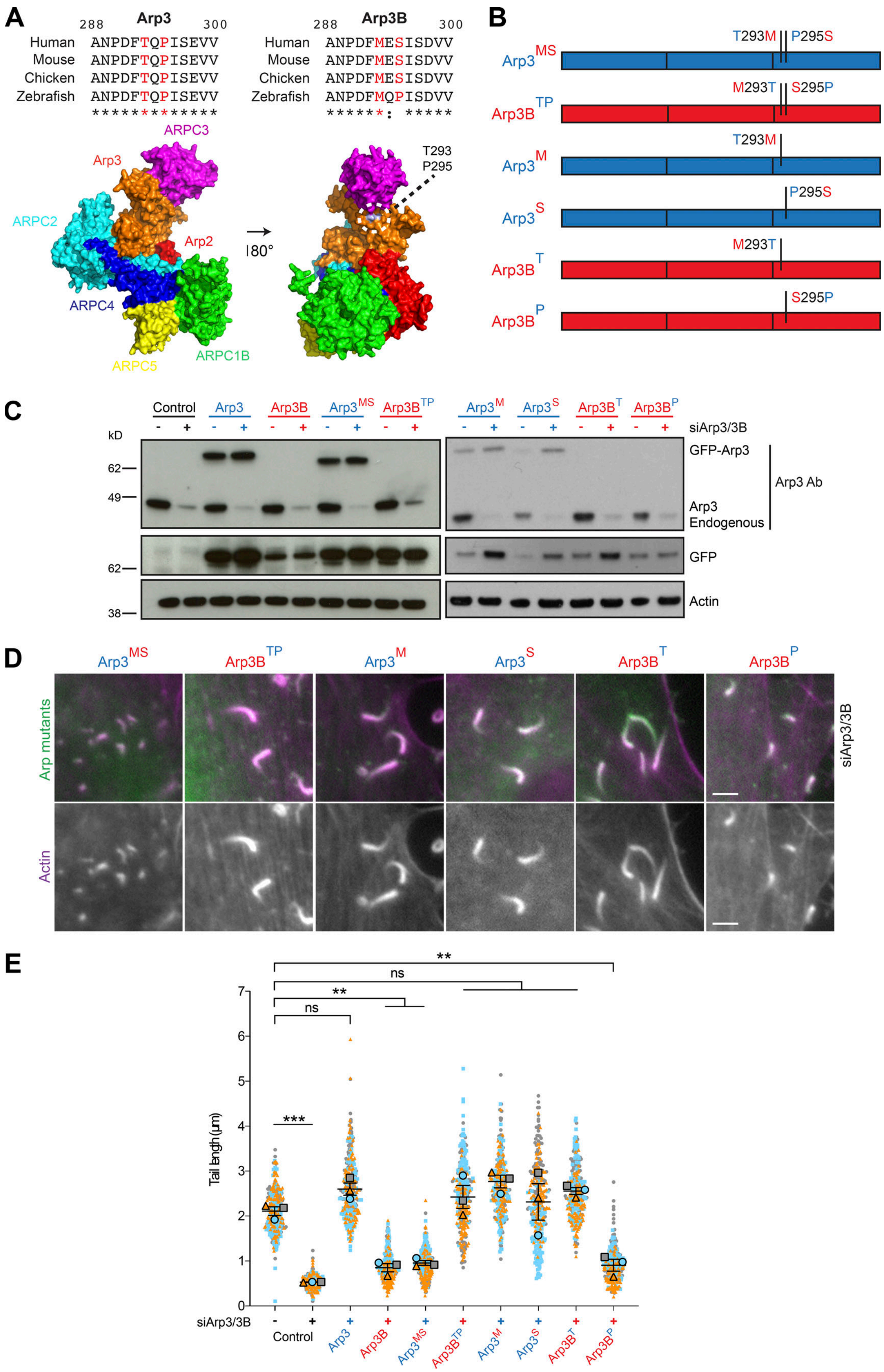


Figure 5. **Met293 of Arp3B is essential to induce short actin tails. (A)** Alignment of residues 288–300 of Arp3 and Arp3B from the indicated species. Conserved residues are indicated with asterisks, and the colon highlights residues with similar properties. Residues 293 and 295 are shown in red. The structure

highlights the position of Thr293 and Pro295 on the surface of Arp3 in the bovine Arp2/3 complex (Protein Data Bank accession no. 1K8K; Robinson et al., 2001). **(B)** Schematic representation of the residue 293 and 295 exchanges between Arp3 (blue) and Arp3B (red) with the substituted amino acids retaining the color of their original protein. **(C)** Immunoblot analysis of the expression level of the indicated RNAi-resistant GFP-tagged Arp3/3B mutants in HeLa cells treated with Arp3 and Arp3B siRNA. **(D)** Immunofluorescence images of representative actin tails labeled with Alexa Fluor 568 phalloidin (magenta) in HeLa cells stably expressing the indicated GFP-tagged Arp3/3B mutants (green) and treated with the indicated siRNA. Scale bars = 5 µm. **(E)** Quantification of actin tail length in cells stably expressing RNAi-resistant GFP-tagged protein and treated with Arp3 and Arp3B siRNA. All error bars represent SEM from $n$ = 3 independent experiments in which the length of 100 tails was analyzed per condition. Tukey's multiple comparisons test was used to determine statistical significance; ***, P < 0.001; **, P < 0.01.

The chemical-physical properties of glutamine mimic oxidized methionine (Met-SO), as the polar double-bonded oxygen is in the same position of the amino acid side chain and the glutamine amide group mimics the low reactivity of the methyl group that terminates the side chain of Met-SO (Aledo, 2019; Balog et al., 2003; Bigelow and Squier, 2011; Drazic et al., 2013; Lundquist et al., 2014). The substitution of Thr293 in Arp3 with glutamine is sufficient to induce an Arp3B phenotype (short actin tails). This suggests that the molecular basis by which this modification promotes debranching is common to all Arp2/3 complexes. Recent cryo-EM structures of DIP1-activated Arp2/3 as well as ARP2/3 at branch points in NIH-3T3 fibroblasts reveals that the residues equivalent to Met293 of Arp3B are in a flexible surface-exposed loop between the ArpC3 and Arp2 subunits (Fäßler et al., 2020; Shaaban et al., 2020). It is not immediately obvious from these structures why oxidation of Met293 or the substitution of a glutamine at this position would promote debranching. However, these structures lack cortactin, which is required for the faster actin disassembly induced by Arp3B-containing complexes, in spite of not being involved in recruitment of coronin 1C or MICAL2. The structure of Arp2/3 at branch points with cortactin will undoubtedly be informative, but a full molecular understanding of MICAL2-mediated debranching is going to require multiple structures, including transition states with bound MICAL2 and coronin.

### MICAL2-induced network disassembly is dependent on coronin and cortactin

It is well established that coronin promotes the disassembly of branched actin networks via a number of different mechanisms (Brieher et al., 2006; Cai et al., 2008; Cai et al., 2007). Coronin binds to actin filaments to enhance severing by altering their conformation and promoting the cooperative recruitment of cofilin and AIP1 (Jansen et al., 2015; Kueh et al., 2008; Mikati et al., 2015; Tang et al., 2020). Coronin 1B, which is complexed with coronin 1C (Abella et al., 2016), also recruits the slingshot phosphatase to lamellipodia, where it is thought to dephosphorylate cofilin to enhance its activity (Cai et al., 2007). In addition, coronin 1B antagonizes cortactin to induce disassembly of actin networks by promoting the debranching of Arp2/3 from the mother filament (Cai et al., 2008). Interestingly, however, in vitro assays with recombinant proteins demonstrate that while coronin 1B preferentially localizes to actin branches, it rarely promotes their debranching when it binds (Cai et al., 2008). Our analysis of the impact of Arp3B on the properties of Arp2/3 complexes has now uncovered an additional level of regulation that increases the rate of coronin 1B/C–mediated actin debranching in cells. Our study also suggests that the relative levels of Arp3 and Arp3B will influence the stability of Arp2/3-induced branch actin networks in different tissues and cell types. Further work will determine whether higher Arp3B expression levels are indeed indicative of a more dynamic actin cytoskeleton. The hierarchical relationship between Arp3B and ARPC1/ARPC5 isoforms in controlling branched actin network dynamics also remains to be established.

In summary, our analysis of the role of ARP2/3 isocomplexes in actin-based motility of vaccinia virus demonstrates how the regulation of Arp3B-containing Arp2/3 complexes by MICAL2, coronin 1C, and cortactin, together with the impact of ARPC1 and ARPC5 isoforms (Abella et al., 2016), can act to fine-tune branched actin network dynamics for different cellular contexts and processes.

## Materials and methods

### Vaccinia virus infection, antibodies, immunoblot, and immunofluorescence analysis

HeLa cells were washed with PBS and then once in serum-free MEM before the virus was added to the cells in serum-free MEM. After 1 h at 37°C, the serum-free MEM was removed and replaced with complete MEM. Cells were incubated at 37°C and fixed or imaged at 8 h after infection. For knockdown experiments, cells were infected with vaccinia virus 72 h after siRNA transfection, and samples from each siRNA condition were kept for quantitative PCR (qPCR) or immunoblot analysis. Cells were fixed with 4% paraformaldehyde in PBS for 10 min, blocked in cytoskeletal buffer (1 mM MES, 15 mM NaCl, 0.5 mM EGTA, 0.5 mM $MgCl_2$, and 0.5 mM glucose, pH 6.1) containing 2% (vol/vol) fetal calf serum and 1% (wt/vol) BSA for 30 min, and then permeabilized with 0.1% Triton/PBS for 5 min. Before permeabilization of the cells with detergent, extracellular enveloped virions (CEV) associated with the plasma membrane were labeled with a monoclonal antibody against B5 (19C2, rat, 1:500 in cytoskeletal buffer; Hiller and Weber, 1985) followed by an Alexa Fluor 647 anti-rat secondary antibody (Invitrogen; 1:500 in PBS). Actin tails were labeled with Alexa Fluor 488 or Alexa Fluor 568 phalloidin (Invitrogen; 1:500 in cytoskeletal buffer). Coverslips were mounted on glass slides using Mowiol (Sigma) and imaged on a Zeiss Axioplan2 microscope equipped a with a 63×/1.4 NA Plan-Achromat objective and a Photometrics Cool Snap HQ cooled charge-coupled device camera. The microscope was controlled with MetaMorph 7.8.13.0 software. Actin tail length was quantified manually using the line tool of Fiji.

The other antibodies used in this study were ARPC2/p34-Arc rabbit polyclonal (Millipore; 07–227, 1:1,000 dilution), ARPC5

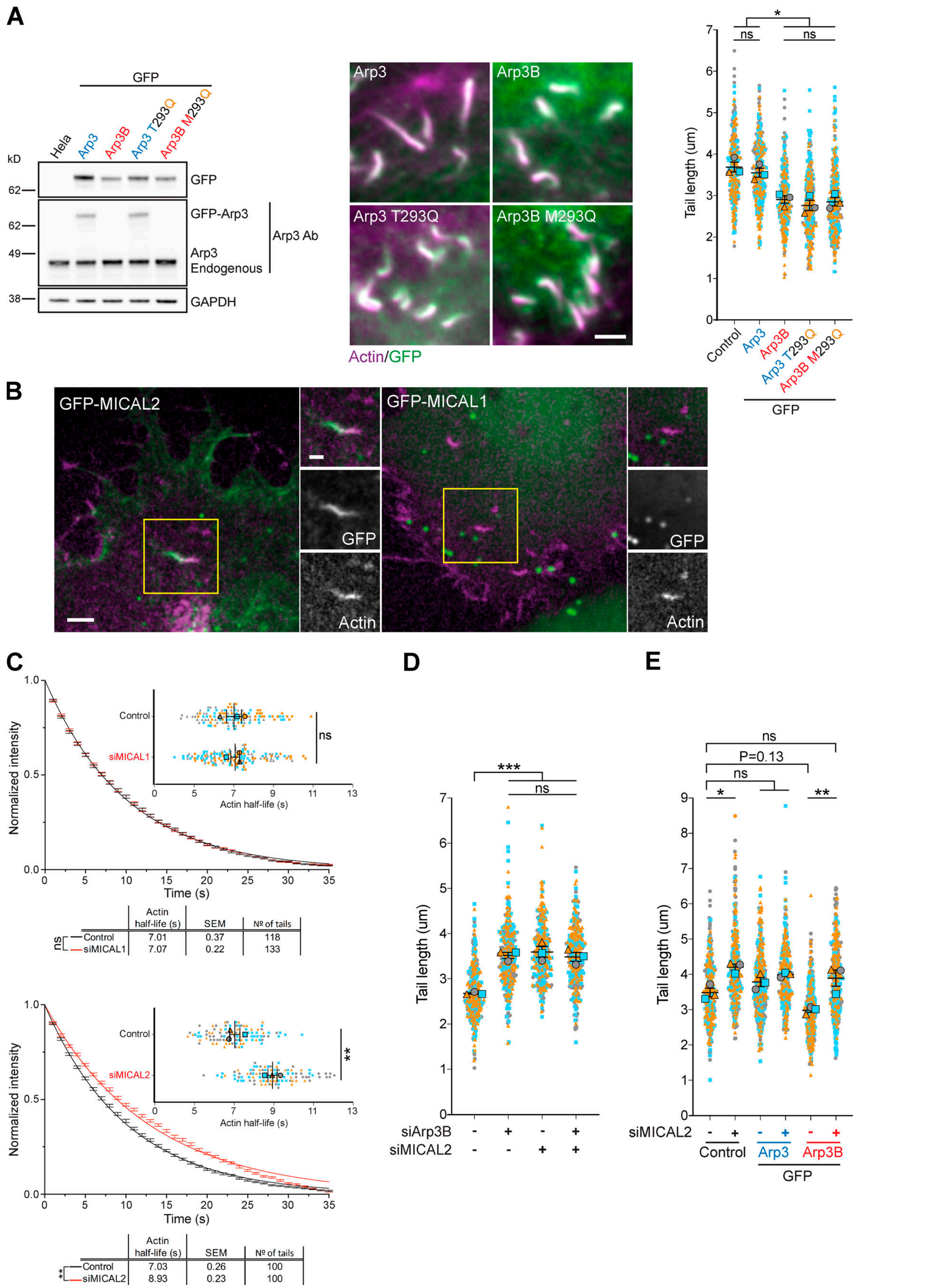


Figure 6. **The ability of Arp3B to induce short actin tails depends on MICAL2. (A)** The immunoblot shows the expression level of GFP-tagged Arp3 (blue) and Arp3B (red) together with their respective glutamine mutants (yellow). The immunofluorescence images show representative actin tails (magenta) in HeLa

cells stably expressing the indicated GFP-tagged proteins (green), while the graph shows the quantification of their length. Scale bar = 2.5 µm. Ab, antibody. **(B)** Image stills from Video 6 and Video 7 showing that GFP-tagged (green) MICAL2, but not MICAL1, is recruited to vaccinia-induced actin tails visualized with LifeAct-iRFP (magenta). Scale bars = 5 µm (main image) and 2.5 µm (insert). **(C)** Quantification of the half-life of photoactivated Cherry-GFP$^{PA}$-β-actin in actin tails in cells treated with MICAL1 and MICAL2 siRNA. The main graphs represent the best-fitting curve for each condition (continuous line) together with the average normalized intensity of the GFP signal at every time point (error bars represent the SEM for the indicated number of tails). The insert graphs show the mean half-life and SEM from $n$ = 3 independent experiments. Student's $t$ test was used to determine statistical significance. **(D)** The graph shows the length of actin tails in HeLa cells treated with the indicated siRNA. **(E)** Quantification of the effect of the loss of MICAL2 on actin tail length in HeLa cells stably expressing GFP-tagged Arp3 and Arp3B. In all actin tail length graphs, the error bars represent SEM from $n$ = 3 independent experiments in which the length of 100 tails was analyzed per condition. Tukey's multiple comparisons test was used to determine statistical significance; ***, P < 0.001; **, P < 0.01; *, P < 0.05.

mouse monoclonal (Synaptic Systems; 305011, clone 323H3, 1:1,000 dilution), Arp2 rabbit monoclonal (Abcam; ab129018, clone EPR7979, 1:1,000 dilution), ARPC5L rabbit monoclonal (Abcam; ab169763, clone EPR10274, 1:1,000 dilution), ARPC1A rabbit polyclonal (Sigma; HPA004334, 1:250 dilution), Arp3 mouse monoclonal (Sigma; A5979, clone FMS338, 1:1000 dilution), ARPC1B rabbit polyclonal (Bethyl Laboratories; A302-781, 1:1,000 dilution), ARPC3 mouse monoclonal (BD Transduction Laboratories; 612234, clone 26/p21-Arc, 1:1,000 dilution), ARPC4 rabbit polyclonal (Sigma; SAB1100901, 1:1,000 dilution), β-actin

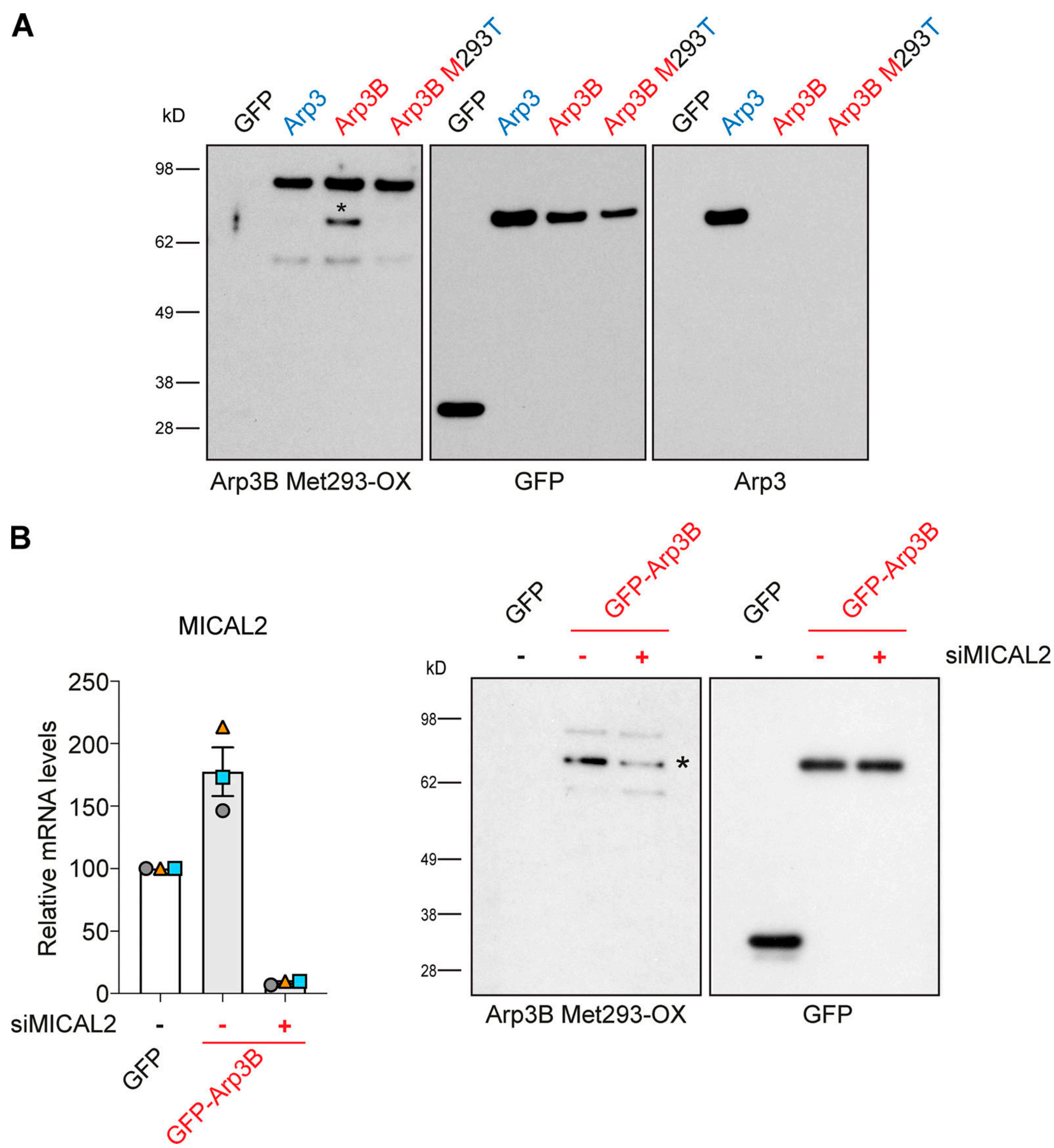


Figure 7. **Arp3B Met293 is oxidized by MICAL2. (A)** Immunoblot analysis of GFP-Trap pulldowns from HeLa cells lysates stably expressing GFP or the indicated GFP-tagged proteins demonstrates that Arp3B (red), but not Arp3 (blue), is oxidized (*) in a Met293-dependent fashion. **(B)** RT-PCR analysis of the level of MICAL2 mRNA in HeLa cells stably expressing GFP or GFP-Arp3B and treated with MICAL2 siRNA. Error bars represent SEM from three independent experiments. The immunoblot of GFP-Trap pulldown assays on cells stably expressing GFP-Arp3B shows the level of Arp3B Met293 oxidation after treatment with MICAL2 siRNA.

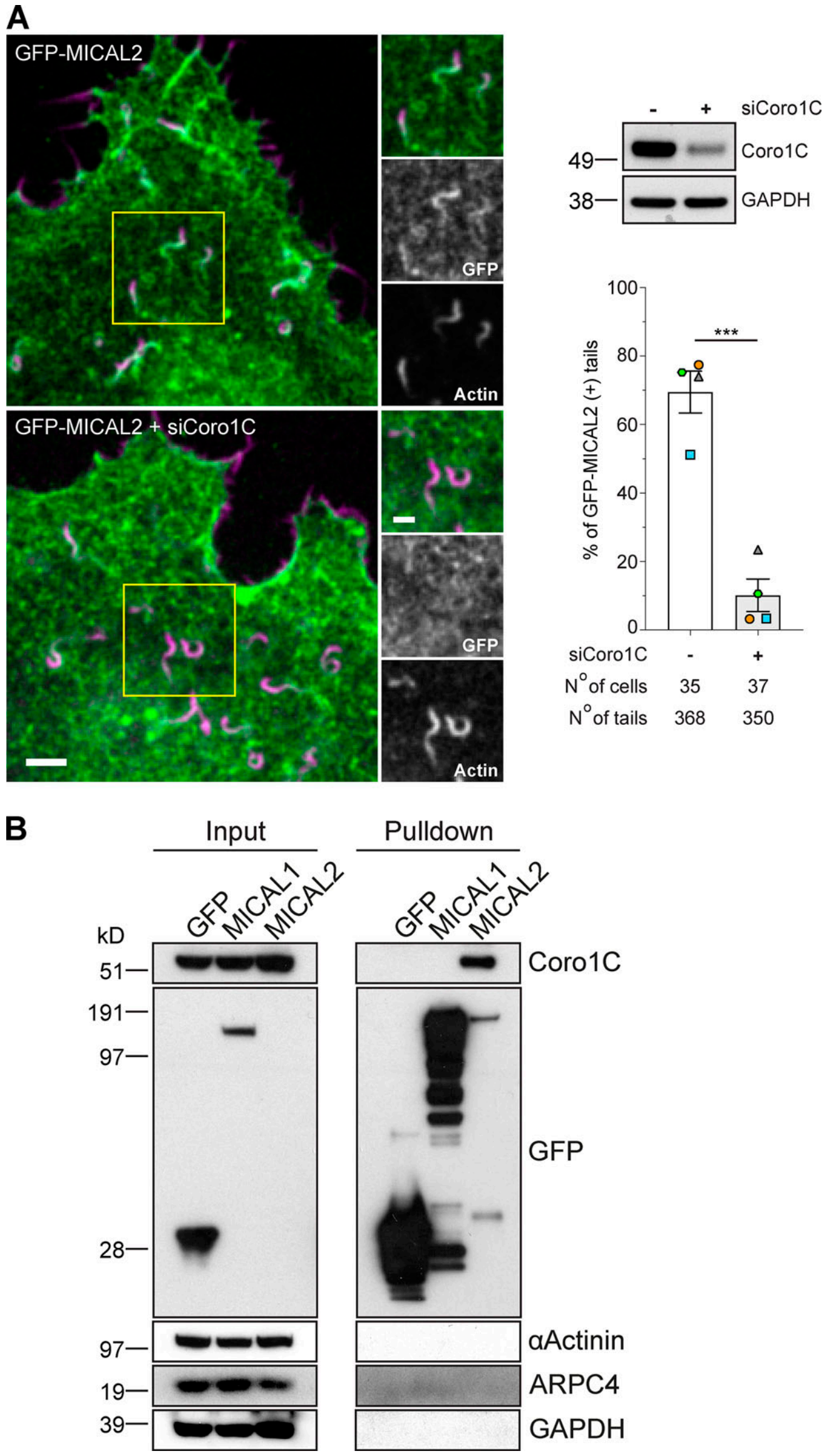


Figure 8. **Coronin 1C recruits MICAL2 to vaccinia-induced actin tails.** **(A)** Image stills from Video 8 and Video 9 showing coronin 1C is required to recruit GFP-MICAL2 (green) to vaccinia-induced actin tails visualized with LifeAct-iRFP (magenta). Scale bars = 5 µm (main image) and 2.5 µm (insert). The immunoblot shows the level of coronin 1C in HeLa cells treated with coronin 1C siRNA, while the graph shows the percentage of actin tails recruiting GFP-MICAL2 following coronin 1C knockdown. The number of cells and actin tails analyzed is indicated. Student's *t* test was used to determine statistical significance for $n$ = 4 independent experiments. **(B)** Immunoblot analysis of GFP-Trap pulldowns on HeLa cells stably expressing the indicated GFP-tagged protein demonstrates coronin 1C (Coro1C) associates with MICAL2, but not MICAL1.

mouse monoclonal (Sigma; A5316, clone AC-74, 1:1,000 dilution), Vinculin mouse monoclonal (Sigma; V4505, 1:1000 dilution), coronin 1C rabbit polyclonal (Invitrogen; PA5-21775, 1:500 dilution), α-Actinin mouse monoclonal (Sigma; A5044, 1:1,000 dilution), GAPDH mouse monoclonal (Santa Cruz Biotechnology; sc-32233, 1:1,000 dilution), cortactin mouse monoclonal (Millipore; 05–180, clone 4F11, 1:1,000 dilution), MICAL1 rabbit polyclonal (Proteintech; 14818-1-AP, 1:1,000 dilution), GFP mouse monoclonal (custom made by Cancer Research UK; 1:1,000 dilution). HRP-conjugated secondary antibodies used in immunoblot analysis were purchased from The Jackson Laboratory.

A peptide corresponding to residues 289–298 of Arp3B with -an additional N-terminal cysteine and Met293 oxidized to Met-SO (CNPDFM(O)ESISD) was synthesized by the peptide facility at the Francis Crick Institute. This peptide used to immunize two rabbits over a 77-d protocol by White Antibodies (www.whiteantibodies.com).

### Stable cell lines

The Cherry-GFP$^{PA}$-β-actin and GFP-coronin 1C stable HeLa cells were previously described (Abella et al., 2016). Stable HeLa cell lines expressing siRES GFP-Arp3, GFP-Arp3B, GFP-Ch1, GFP-Ch2, GFP-Ch3, GFP-Arp3$^{MS}$, GFP-Arp3B$^{TP}$, GFP-Arp3$^{M}$, GFP-Arp3$^{S}$, GFP-Arp3B$^{T}$, GFP-Arp3B$^{P}$, GFP$^{PA}$-Arp3, GFP$^{PA}$-Arp3B, GFP-MICAL1 and GFP-MICAL2 were generated using the lentivirus infection (Trono group second generation packaging system, Addgene) and selected using puromycin resistance (1 µg/ml) as previously described (Abella et al., 2016). For experiments analyzing actin tail length in cell lines listed above, parental HeLa cells were used as internal control (indicated as "control" in figures). The siRES GFP$^{PA}$-Arp3 or GFP$^{PA}$-Arp3B lentiviral constructs were used to infect LifeAct-RFP stable HeLa cells (Abella et al., 2016), and cells expressing both proteins were selected using a combination of hygromycin and puromycin (100 µg/ml and 1 µg/ml, respectively).

### Expression constructs

The expression vectors pLVX–LifeAct–RFP, pLVX–GFP$^{pa}$–Cherry-β-actin, and pLVX–ARPC2-GFP have been previously described (Abella et al., 2016). Human Arp3, Arp3B, chimeras 1–3, Arp3$^{MS}$, Arp3B$^{TP}$, and the single point mutants were obtained as synthetic genes (Invitrogen; Geneart) and cloned into the NotI/EcoRI sites of pLVX–N-term–GFP (Abella et al., 2016). Using pLVX-GFP-Arp3 and pLVX-GFP-Arp3B as a template, primers containing a mismatch for the T293Q and M293Q mutation were used to amplify two overlapping fragments corresponding to Arp3 or Arp3B, respectively. These two fragments were then assembled into NotI/EcoRI-digested pLVX–N-term–GFP using Gibson Assembly to obtain pLVX-GFP-Arp3-T293Q and pLVX-GFP-Arp3B-M293Q. Human MICAL1 was obtained by PCR amplification from a GFP-MICAL1 plasmid provided by Arnaud Echard (Institut Pasteur, Paris, France; Frémont et al., 2017a). This PCR fragment was cloned into the NotI/EcoRI sites of pLVX–N-term–GFP to generate pLVX-GFP-human-MICAL1. Human MICAL2 was obtained by PCR amplification from a HA-MICAL2 plasmid donated by Steve Caplan (University of Nebraska, Omaha, NE; Giridharan et al., 2012). This PCR fragment was cloned into the NotI/EcoRI sites of pLVX–N-term–GFP using Gibson Assembly to generate pLVX-GFP-human-MICAL2. LifeAct-iRFP670 was amplified from pLVX-LifeAct-iRFP670 (Snetkov et al., 2016) and cloned using Gibson Assembly into a BglII/BamHI-digested pE/L vector to obtain pE/L-LifeAct-iRFP670.

### Live-cell imaging and photoactivation

All live-cell imaging experiments were performed on HeLa cells in complete MEM (10% FBS) in a temperature-controlled

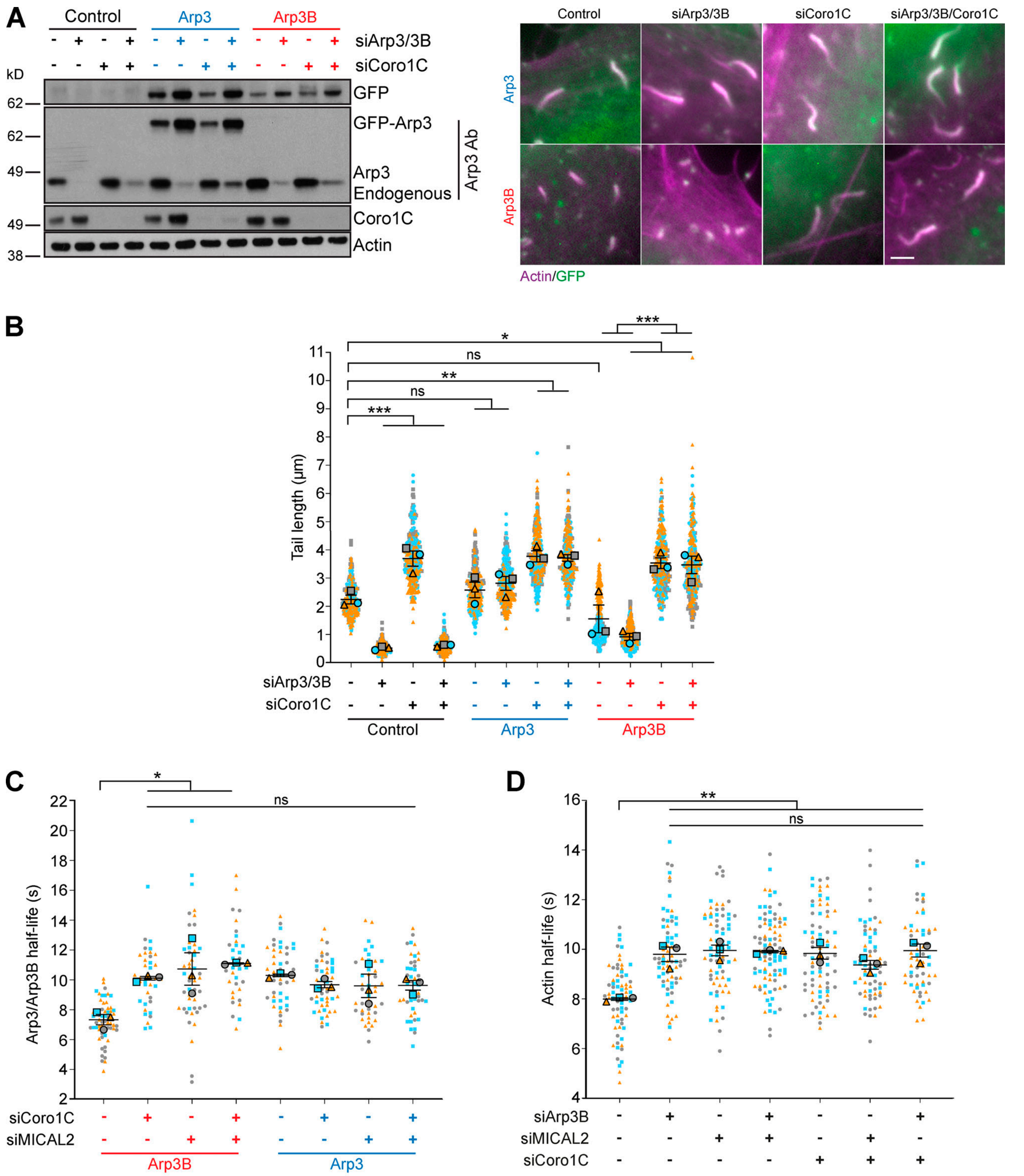


| GFPpa | Half-life (s) | SEM | Nº of tails |
|---|---|---|---|
| Arp3B Control | 7.33 | 0.35 | 62 |
| Arp3B + siCoro1C | 10.11 | 0.13 | 38 |
| Arp3B + siMICAL2 | 10.73 | 1.09 | 49 |
| Arp3B + siCoro1C-MICAL2 | 11.10 | 0.04 | 34 |
| Arp3 Control | 10.30 | 0.09 | 45 |
| Arp3 + siCoro1C | 9.68 | 0.20 | 49 |
| Arp3 + siMICAL2 | 9.60 | 0.78 | 44 |
| Arp3 + siCoro1C-MICAL2 | 9.64 | 0.32 | 48 |

| | Actin half-life (s) | SEM | Nº of tails |
|---|---|---|---|
| Control | 8.00 | 0.05 | 72 |
| siArp3B | 9.80 | 0.29 | 71 |
| siMICAL2 | 9.96 | 0.22 | 71 |
| siArp3B + siMICAL2 | 9.91 | 0.05 | 92 |
| siCoro1C | 9.84 | 0.23 | 76 |
| siCoro1C + siMICAL2 | 9.37 | 0.18 | 69 |
| siArp3B + siCoro1C | 9.95 | 0.26 | 60 |

Figure 9. **Loss of coronin 1C increases stability of Arp3B, but not Arp3 actin networks. (A)** The immunoblot shows the expression level of RNAi-resistant GFP-tagged Arp3 (blue) and Arp3B (red) in HeLa cells treated with Arp3/Arp3B and coronin 1C siRNA. The immunofluorescent images show representative

actin tails (magenta) in the siRNA-treated HeLa cells stably expressing GFP-tagged Arp3 and Arp3B (green). Scale bar = 5 µm. **(B)** Quantification of actin tail lengths in cells stably expressing RNAi-resistant GFP-tagged Arp3 or Arp3B and treated with Arp3/Arp3B and coronin 1C siRNA. The error bars represent SEM from $n$ = 3 independent experiments in which the length of 100 tails was analyzed per condition. **(C)** Quantification of the half-life of photoactivated $GFP^{PA}$-tagged Arp3 or Arp3B in actin tails in cells treated with coronin 1C and/or MICAL2 siRNA. **(D)** Quantification of the half-life of photoactivated Cherry-$GFP^{PA}$-β-actin in actin tails in cells treated with different combinations of Arp3B, coronin 1C, and MICAL2 siRNA. The graphs in C and D show the mean half-life and SEM from $n$ = 3 independent experiments. Tukey's multiple comparisons test was used to determine statistical significance; ***, P < 0.001; **, P < 0.01; *, P < 0.05.

chamber at 37°C. Photoactivation experiments involving $GFP^{PA}$ tags were performed as previously described (Abella et al., 2016). Briefly, cells were infected with YFP-A3 vaccinia virus (Arakawa et al., 2007) and imaged on a Zeiss Axio Observer spinning-disk microscope equipped with a Plan Achromat 63×/1.4 Ph3 M27 oil lens, an Evolve 512 camera, and a Yokagawa CSUX spinning disk. The microscope was controlled by the Slidebook software (3i intelligent imaging innovations). The $GFP^{pa}$ was photoactivated in a region of interest underneath the virus using a focused 1-ms pulse of a 405-nm laser. The decay of the GFP signal was followed over time by imaging every second (1 Hz). The Fiji plugin FRAP analyser, created by D. Barry (Crick Institute), was used to obtain the values of the GFP signal in the regions of interest used for photoactivation. These values were background subtracted and normalized to the first frame after photoactivation. The decay of the photoactivated GFP signal was analyzed by one-phase exponential fitting using GraphPad Prism 8 to obtain the half-lives values for each condition.

To visualize actin structures in HeLa cells stably expressing GFP-MICAL1 and GFP-MICAL2, the cells were transfected with pE/L-LifeAct-iRFP670 using FUGENE (Promega) and infected with vaccinia virus before live imaging. Images were acquired every 3 s for 3 min at 8 h after infection on a Zeiss Axio Observer Microscope using the 63× 1.4 NA objective (see above for microscope configuration). Videos of control and siCoro1C transfected cells were blind scored in Fiji for the presence of GFP-MICAL2 on actin tails. Data are presented as the mean ± SEM of the percentage of GFP-positive tails over four independent experiments.

### siRNA transfection

HeLa cells were transfected with siRNA as previously described (Abella et al., 2016). The following siRNAs were used: All-Star (indicated as control in siRNA experiments; Qiagen; SI03650318), siRNA pools of four oligos against Arp3 (Dharmacon; M-012077-01-0010: 5′-GGAAUAUACUGCUGAAAUA-3′, 5′-GACCUAGACUUCUUCAUUG-3′, 5′-GAACAAUCCUUGGAAACUG-3′, and 5′-UAAAGGAGCGCUAUAGUUA-3′), Arp3B (Dharmacon; M-020372-01-0010: 5′-CCAGAAGAAGUUUGUUAUA-3′, 5′-AAACAGAGAGUAUCUUGCA-3′, 5′-UAACGUACCAGGACUCUAC-3′, and 5′-CGAUAAACCUACAUAUGCU-3′), cortactin (Dharmacon; MU-010508-00-0010: 5′-GAACAAGACCGAAUGGAUA-3′, 5′-GAAUAUCAGUCGAACUUU-3′, 5′-GGACAGAGUUGAUCAGUCU-3′, and 5′-ACAGAGAGAUUACUCCAAA-3′), coronin 1C (Dharmacon; M-017331-00-0010: 5′-GCACAAGACUGGUCGAAUU-3′, 5′-GGGAAGCCCUUAUAAACUU-3′, 5′-GAGCAAGUUUCGGCAUGUA-3′, and 5′-GCAAUCAAGAUGAGCGUAU-3′), and MICAL2 (Dharmacon; L-010189-00-0005: 5′-GGACGAGCCUUCAAACUUU-3′, 5′-GCAUCCACGUGCAAAGGUA-3′, 5′-GCAGGAACGCCGUGUCUCA-3′, and 5′-UAUCAGUAUUCGCCAACUA-3′). A MICAL1-targeting siRNA (5′-GAGUCCACGUCUCCGAUUU-3′) was designed as previously described (Frémont et al., 2017a). Deconvoluted siRNAs were used against Arp3 (Dharmacon; D-012077-01: 5′-GGAAUAUACUGCUGAAAUA-3′, D-012077-03: 5′-GACCUAGACUUCUUCAUUG-3′, D-012077-04: 5′-GAACAAUCCUUGGAAACUG-3′, and D-012077-05: 5′-UAAAGGAGCGCUAUAGUUA-3′) and Arp3B (Dharmacon; D-020372-01: 5′-CCAGAAGAAGUUUGUUAUA-3′, D-020372-02: 5′-AAACAGAGAGUAUCUUGCA-3′, D-020372-03: 5′-UAACGUACCAGGACUCUAC-3′, and D-020372-04: 5′-CGAUAAACCUACAUAUGCU-3′).

### GFP-Trap pulldown assays

Cells were treated with 200 nM Latrunculin A for 1 h and then lysed in the following buffer: 50 mM Hepes, pH 7.5, 150 mM NaCl, 1.5 mM $MgCl_2$, 1 mM EGTA, 1% Triton X-100, 10% glycerol, 50 mM NaF, 1 mM sodium vanadate, 200 nM Latrunculin A, Phosphostop, and protease inhibitors (Roche). Lysates were normalized and incubated with GFP-Trap agarose beads (Chromotek), and pulldown was performed according to the manufacturer's instructions.

### Recombinant expression and purification of Arp2/3 complexes

The multibac expression vector pFL system (Fitzgerald et al., 2006) was used to express all Arp2/3 complexes in Sf21 insect cells. The following vectors were used to produce the corresponding baculoviruses for infection of Sf21 cells: pFL–ARPC2–Arp3, pFL–ARPC2–Arp3B, pFL–ARPC4–ARPC1A–TEV–STREP, pFL–ARPC4–ARPC1B–TEV–STREP, pFL–ARPC3–ACTR2–ARPC5, and pFL–ARPC3–ACTR2–ARPC5L. Human Arp3B was amplified by PCR using pLVX-GFP-Arp3B as a template and cloned into the BamHI/NotI sites of the pFL vector, which also contains ArpC2 (Abella et al., 2016). The baculovirus expression vectors for the other subunits/isoforms were previously described (Abella et al., 2016). The TEV-cleavable double STREP tag at the C terminus of ARPC1A/ARPC1B was used for affinity purification. Isoform-specific Arp2/3 complexes were produced using a combination of three baculoviruses from the list above, depending on the isoform composition. The three viruses were used to coinfect Sf21 insect cells maintained in SF900-III medium (Life Technologies). At 72 h after infection, cells were harvested by centrifugation and resuspended in buffer A (50 mM Tris, pH 8.0, 100 mM KCl, 5 mM EGTA, 1 mM EDTA, 2 mM $MgCl_2$, 0.5 mM DTT, 0.2 mM ATP, 2.5% [vol/vol] glycerol, and protease inhibitor [Roche]). Cells were lysed by sonication, and the insoluble fraction was removed by centrifugation at 80,000 $g$ for 30 min. The resulting soluble fraction was loaded onto a 1 ml StrepTrap HP disposable 1 ml column (GE Healthcare) at 1 ml $min^{-1}$. After washing, the bound fraction was eluted

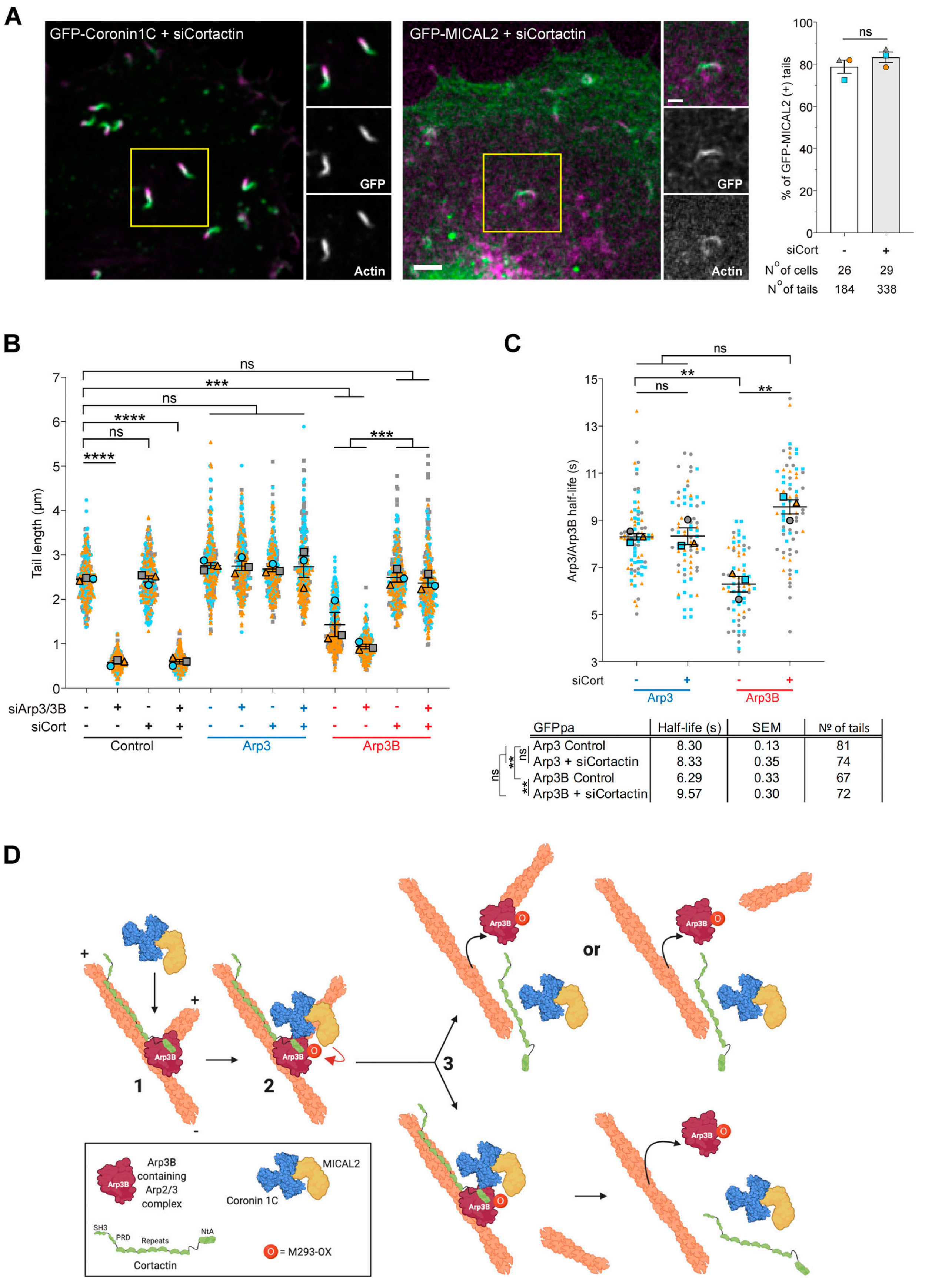


Figure 10. **MICAL2-mediated actin network disassembly depends on cortactin. (A)** Image stills from Video 10 showing GFP-tagged coronin 1C and MICAL2 (green) are still recruited to actin tails visualized with LifeAct-iRFP (magenta) in cells treated with cortactin siRNA. Scale bars = 5 µm (main image) and

2.5 µm (insert). The graph shows the percentage of actin tails recruiting GFP-MICAL2 following cortactin knockdown. The number of cells and actin tails analyzed is indicated. Student's *t* test was used to determine statistical significance for *n* = 3 independent experiments. **(B)** Quantification of actin tail length in HeLa cells stably expressing RNAi-resistant GFP-tagged Arp3 (blue) or Arp3B (red) and treated with Arp3/Arp3B and cortactin siRNA. The error bars represent SEM from *n* = 3 independent experiments in which the length of 100 tails were analyzed per condition. **(C)** Quantification of the half-life of photoactivated $GFP^{PA}$-tagged Arp3 or Arp3B in actin tails in cells treated with cortactin siRNA. The graph shows the mean half-life and SEM from *n* = 3 independent experiments. Tukey's multiple comparisons test was used to determine statistical significance; ****, P < 0.0001; ***, P < 0.001; **, P < 0.01. **(D)** The schematic illustrates how MICAL2, coronin 1C, and cortactin cooperate to promote actin filament debranching by oxidizing Met293 of Arp3B. The plus and minus ends of the actin filaments are indicated. The MICAL2–coronin 1C complex is recruited to actin branches (step 1). MICAL2 oxidizes Met293 on Arp3B (step 2). Cortactin is required to facilitate ARP3B oxidation and/or promote subsequent debranching (step 3). It remains to be established if the daughter filament dissociates before the Arp2/3 complex (bottom row) or whether both dissociate at the same time, possibly with the complex attached to the end of the daughter filament (top row). NtA, amino-terminal acidic region; PRD, proline-rich domina; SH3, SRC homology 3 domain.

in buffer A with 10 mM D-Desthiobiotin (Sigma; D1411). The eluted Arp2/3 complex was diluted 1:2 in the Mono S dilution buffer (20 mM MES, pH 6.1, 2 mM $MgCl_2$, 5 mM EGTA, 1 mM EDTA, 0.5 mM DTT, 0.2 mM ATP, and 10% [vol/vol] glycerol) and loaded onto a cation exchange Mono S 5/50 GL column (GE Healthcare) at 1 ml $min^{-1}$. Buffer B (20 mM MES, pH 6.1, 2 mM $MgCl_2$, 50 mM KCl, 5 mM EGTA, 1 mM EDTA, 0.5 mM DTT, 0.2 mM ATP, and 10% [vol/vol] glycerol) was used as running buffer, and elution was performed with buffer C (20 mM MES, pH 6.1, 2 mM $MgCl_2$, 500 mM KCl, 5 mM EGTA, 1 mM EDTA, 0.5 mM DTT, 0.2 mM ATP, and 10% [vol/vol] glycerol). The complexes eluted around 200 mM KCl. The eluted fractions containing all seven subunits were pooled and further purified by size exclusion on a Superose12 (10/300) column and buffer D (20 mM MOPS, pH 7.0, 100 mM KCl, 2 mM $MgCl_2$, 5 mM EGTA, 1 mM EDTA, 0.5 mM DTT, and 0.2 mM ATP, 5% [vol/vol]) as the final buffer. Aliquots of purified complexes were flash frozen in liquid nitrogen and stored at −80°C.

### Pyrene actin polymerization, TIRF branching assays, and analysis

Pyrene-labeled and unlabeled actin monomers were mixed in G-buffer (2 mM Tris, pH 8.0, 0.2 mM ATP, 0.1 mM $CaCl_2$, and 0.5 mM DTT) to obtain the desired ratio of pyrene actin to unlabeled actin (10% pyrene actin) and diluted to 20 µM total actin in G buffer. Actin polymerization was performed in presence of 2 µM monomeric actin (10% pyrene labeled) at room temperature by the addition of one-tenth volume of 10× KMEI (5 mM Tris-HCl, pH 8.0, 0.2 mM ATP, 0.1 mM $CaCl_2$, and 0.5 mM DTT) in G buffer. Experiments were conducted in the presence or in the absence of different Arp2/3 complexes at three different concentrations (12.5, 25, and 50 nM). The polymerization of actin was followed by changes in pyrene fluorescence using excitation and emission wavelengths of 365 and 407 nm, respectively, on a Xenius SAFAS fluorimeter (SAFAS SA). The data were analyzed with GraphPad Prism 6.

The time course of actin assembly (0.6 µM) in the presence of 0.5, 2.5, or 5.0 nM Arp2/3 complex and 50 nM GST (human WASp-VCA) was visualized using an objective-based azimuthal ilas2 TIRF microscope (Nikon Eclipse Ti, modified by Roper Scientific) equipped with a CFI Apochromat TIRF 60×C oil ×60 NA 1.49 TIRF objective and an Evolve 512 camera (Photometrics). Excitation was achieved using lasers with wavelengths of 491 and 561 nm (Optical Insights). Time-lapse recording was performed using MetaMorph software (Universal Imaging; version 7.7.5) at one image every 10 s. Automated quantification of TIRF actin assembly assays was performed with the AnaMorf ImageJ plugin (Barry et al., 2015; details are available at https://github.com/djpbarry/AnaMorf/wiki).

### RNA extraction and qPCR

Purification of total RNA from cells was performed using the QIAshredder and RNeasy Mini kit (Qiagen) according to the manufacturer's instructions. Reverse transcription reactions were performed using SuperScript II Reverse Transcription and RNaseOUT Recombinant Ribonuclease Inhibitor (Thermo Fisher Scientific) to obtain cDNA. qPCR analysis was performed using Power UP SYBR Green Master Mix (Thermo Fisher Scientific) and 7500 Fast Real-Time PCR System (Applied Biosystems). The mRNA levels of the genes of interest were normalized using GAPDH mRNA. Normalized mRNA levels were compared between siRNA-treated and control conditions using the comparative Ct method. The following primers were used: GAPDH forward 5′-TGATGACATCAAGAAGGTGGTG-3′, reverse 5′-TCCTTGGAGGCCATGTGGGCCA-3′; ACTR3 forward 5′-CGATATGCAGTTTGGTTTGG-3′, reverse 5′-TTTGGTGTGGCATACTTGGT-3′; ACTR3B forward 5′-GCCCGCTGTATAAGAATGTCG-3′, reverse 5′-AATCCCTGAACATGGTGGAGC-3′; MICAL2 forward 5′-GTGCACGAACACCAAGTGTC-3′, reverse 5′-CCAGAGGTGTAGCACGTTGT-3′; and coronin 1C forward 5′-GGTTTCTCGTGTGACCTGGG-3′, reverse 5′-TCAATTCGACCAGTCTTGTGC-3′.

### Statistical analysis

For actin tail length experiments, 10 cells per condition were acquired, and 10 tails per cell were measured in each independent experiment (100 tails per condition in total). For actin tail length and protein turnover experiments, the means of the three independent experiments were used to determine statistical significance. When only two conditions were compared, two-sided Student *t* test was performed to determine statistical significance. When more than two samples were compared, Tukey's multiple comparisons test was used to determine statistical significance. All data are represented as super plots to allow assessment of the data distribution in individual experiments (Lord et al., 2020). Data distribution was assumed to be normal, but this was not formally tested. All data were analyzed using GraphPad Prism 8.

### Online supplemental material

Fig. S1 shows the qPCR and immunoblot analysis of experiments in Figs. 1, 2, and 3. Fig. S2, related to Fig. 5, shows the position of

residues 293 and 295 in two different structures of Arp2/3 in its active conformation. Fig. S3 shows the qPCR and immunoblot analysis as well as immunofluorescence images of experiments in Fig. 6. Fig. S4 shows qPCR analysis and exponential curve fitting of photoactivation experiments in Fig. 9. Fig. S5 shows the qPCR, immunoblot analysis, immunofluorescence images, and exponential curve fitting of experiments in Fig. 10. Video 1, related to Fig. 2, shows in vitro TIRF assays of branched actin network nucleation by Arp3- and Arp3B-containing complexes. Video 2 and Video 3 show photoactivation of mCherry-GFP$^{pa}$-actin in tails after treatment with control or Arp3B siRNAs, respectively (related to Fig. 3, A and B). Video 4 and Video 5 show photoactivation of GFP$^{PA}$-Arp3 and GFP$^{PA}$-Arp3B, respectively, in actin tails (related to Fig. 3, C and D). Video 6 and Video 7 show the localization of GFP-MICAL2 and GFP-MICAL1, respectively, in infected cells (related to Fig. 6 B). Video 8 and Video 9 show the localization of GFP-MICAL2 in infected cells after treatment with control or coronin 1C siRNAs, respectively (related to Fig. 8). Video 10 shows that the recruitment of GFP-Coro1C and GFP-MICAL2 to actin tails is not affected by depletion of cortactin (related to Fig. 10 A).

## Acknowledgments

We would like to thank Arnaud Echard (Institut Pasteur, Paris, France) and Steve Caplan (University of Nebraska, Omaha, NE) for providing GFP-MICAL1 and HA-MICAL2 plasmids, respectively, and Nicola O'Reilly for synthesis of the Arp3B MET293-OX peptide (Peptide Chemistry, Francis Crick Institute). We thank Joseph Cockburn (Leeds University) and Carolyn Moores (Birkbeck College, London) for structural insights, as well as Alessio Yang and Angika Basant from the Way laboratory for providing the pE/L-LifeAct-iRFP670 expression vector. We would like to thank Richard Treisman and Angika Basant (Francis Crick Institute), Carolyn Moores (Birkbeck College, London), and Edgar Gomes (Instituto de Medicina Molecular, Lisbon, Portugal) for critically reading the manuscript and scientific discussion.

M. Way was supported by Cancer Research UK (FC001209), UK Medical Research Council (FC001209), and Wellcome Trust (FC001209) funding at the Francis Crick Institute. The project also received funding from the European Research Council under the European Union's Horizon 2020 research and innovation programme (grant 810207 to M. Way) and the European Research Council European Research AAA (grant 741773 to L. Blanchoin).

The authors declare no competing financial interests.

Author contributions: C. Galloni, D. Carra, and N. Kogata generated stable HeLa cell lines (GFP-tagged Arp3, Arp3B, Ch1, Ch2, Ch3, Arp3$^{MS}$, Arp3B$^{TP}$, Arp3$^{M}$, Arp3$^{S}$, Arp3B$^{T}$, and Arp3B$^{P}$; GFP-tagged Arp3-T293Q, Arp3B-M293Q, MICAL1, and MICAL2; and GFP$^{PA}$-Arp3 and GFP$^{PA}$-Arp3B, respectively). Actin tail lengths were quantified by C. Galloni (Figs. 1, S1, 4, 5, 9, and 10) and D. Carra (Fig. 6). Immunoblots and RT-PCR were generated by C. Galloni (Fig. 1; Fig. S1, A–D; and Figs. 4 C, 5 C, S5 B, and 9 A) and D. Carra (Fig. S1, E and F; Figs. S3 and S4; Fig. S5, A and D; and Figs. 6 A, 7, and 8). GFP-TRAP pulldowns in Fig. 1 B and Figs. 7 and 8 B were performed by C. Galloni and D. Carra, respectively. All photoactivation experiments (Figs. 3, S4, S5, 6, 9, and 10) and their analyses were performed by D. Carra, as were GFP-MICAL and GFP-coronin 1C localization experiments (Figs. 6 B, 8 A, and 10 A). S. Kjær expressed and purified recombinant Arp2/3 complexes. P. Singaravelu, C. Guérin, and L. Blanchoin performed actin assembly assays in Fig. 2 and provided scientific discussion. D.J. Barry analysed TIRF assays and provided advice concerning analysis of photoactivation experiments. J.G.V. Abella and M. Way conceived and supervised the work.

Submitted: 6 February 2021

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

## Supplemental material

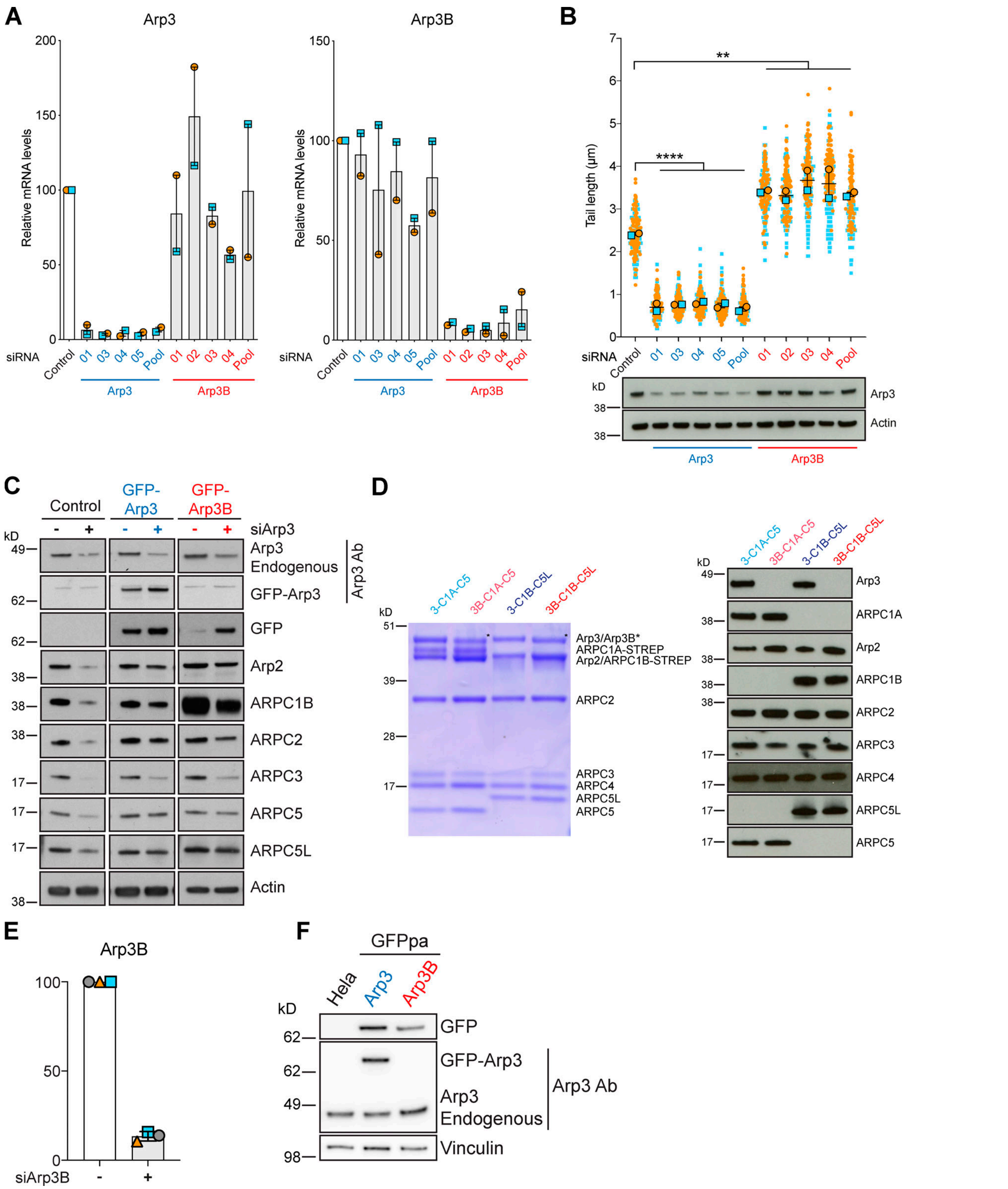


Figure S1. **Arp3 and Arp3B knockdown and rescue. (A)** RT-PCR analysis showing the level of Arp3 (blue) and Arp3B (red) mRNA relative to control in HeLa cells after knockdown with the individual siRNA from the siGenome pools against Arp3 and Arp3B. Error bars represent SEM from two independent experiments. **(B)** Quantification of actin tail lengths and immunoblot analysis of the level of Arp3 in HeLa cells treated with the indicated siRNA. The error bars represent SEM from *n* = 2 independent experiments in which the length of 100 tails was analyzed per condition. Tukey's multiple comparisons test was used to determine statistical significance; ****, $P < 0.0001$; **, $P < 0.01$. **(C)** Immunoblot analysis showing that stable expression of RNAi-resistant GFP-tagged Arp3 (blue) and Arp3B (red) restore control levels of all Arp2/3 subunits in HeLa cells treated with Arp3 siRNA. **(D)** Coomassie-stained gel of recombinant isoform-specific Arp2/3 complexes and corresponding immunoblot. **(E)** RT-PCR analysis of the level of Arp3B mRNA in HeLa cells stably expressing Cherry-GFP$^{PA}$-β-actin and treated with Arp3B siRNA. Error bars represent SEM from three independent experiments. **(F)** Immunoblot analysis of lysates from HeLa cells stably expressing GFP$^{PA}$-tagged Arp3 (blue) and Arp3B (red).

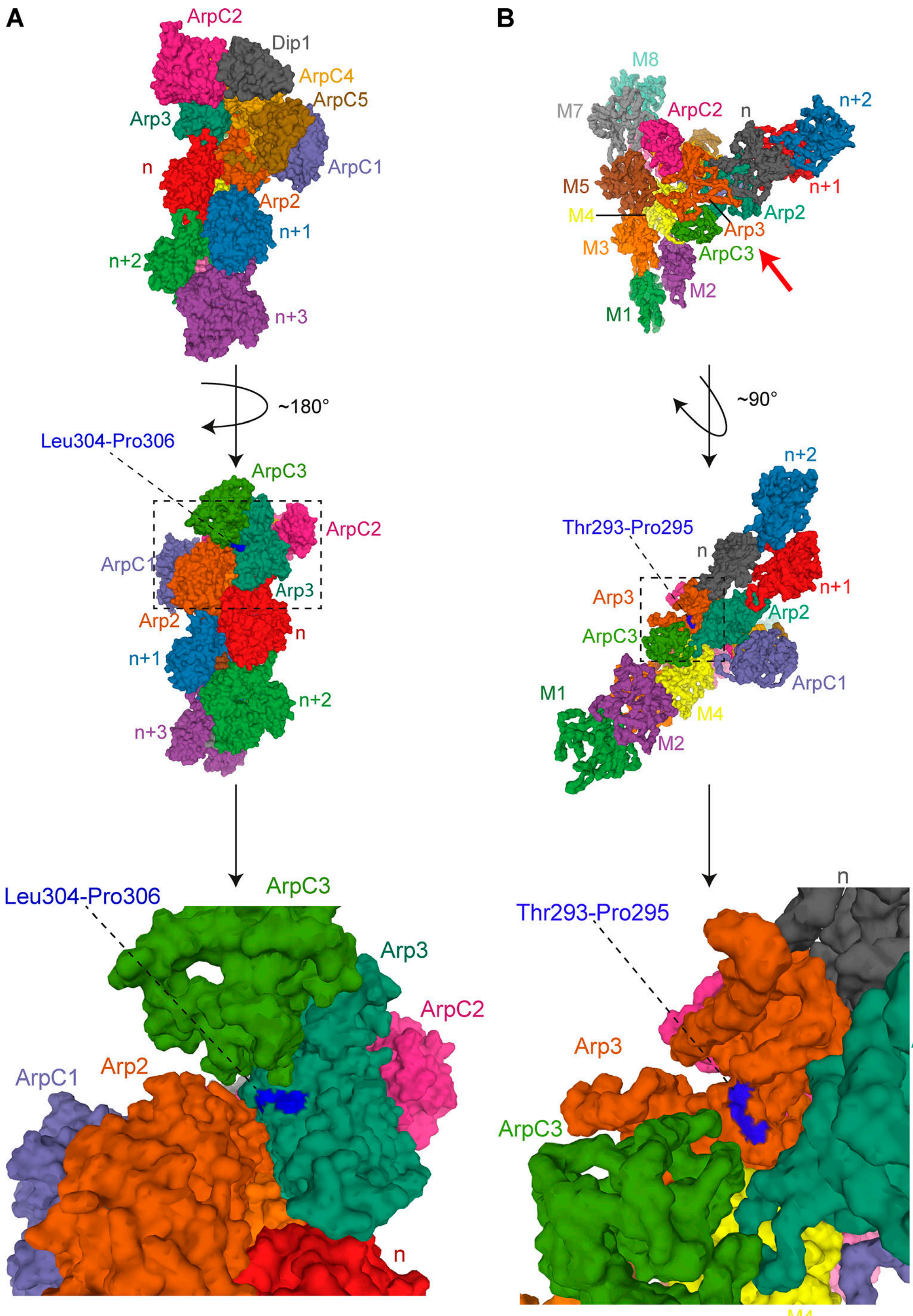


Figure S2. **Residues 293/295 are surface exposed in activated Arp2/3. (A)** Top: 3.9-Å cryo-EM structure of *S. pombe* Arp2/3 in its short-pitch conformation after activation by Dip1 (Shaaban et al., 2020; Protein Data Bank accession no. 6W17). The structure is visualized as molecular surface, and the Arp2/3 subunits, Dip1, and the daughter actin filament subunits (indicated as n, n+1…etc) are color coded as in Protein Data Bank accession no. 6W17. Middle: The structure is rotated ∼180° in the direction indicated by the black curved arrow to allow visualization of residues of Leu304 and Pro306 of *Schizosaccharomyces pombe* Arp3 (highlighted in dark blue) that correspond to Met293 and Ser295 in human Arp3B. The dashed rectangle is enlarged at the bottom of the figure. **(B)** Top: 9.0-Å cryo-EM structure of human Arp2/3 at the branch point between a mother and daughter filament (Fäßler et al., 2020; Protein Data Bank accession no. 7AQK). The structure is visualized as molecular surface, and the Arp2/3 subunits, mother filament subunits (indicated as M1–M8), and daughter filament subunits (indicated as n, n + 1…etc.) are color coded as in Protein Data Bank accession no. 7AQK. The red arrow below the branch point indicates the point of view represented in the middle panel after a ∼90° rotation in the direction shown by the black curved arrow. This structure contains Arp3; hence, the position of Thr293 and Pro295 is highlighted in dark blue and indicated by a dashed black line. The dashed rectangle is enlarged at the bottom of the figure.

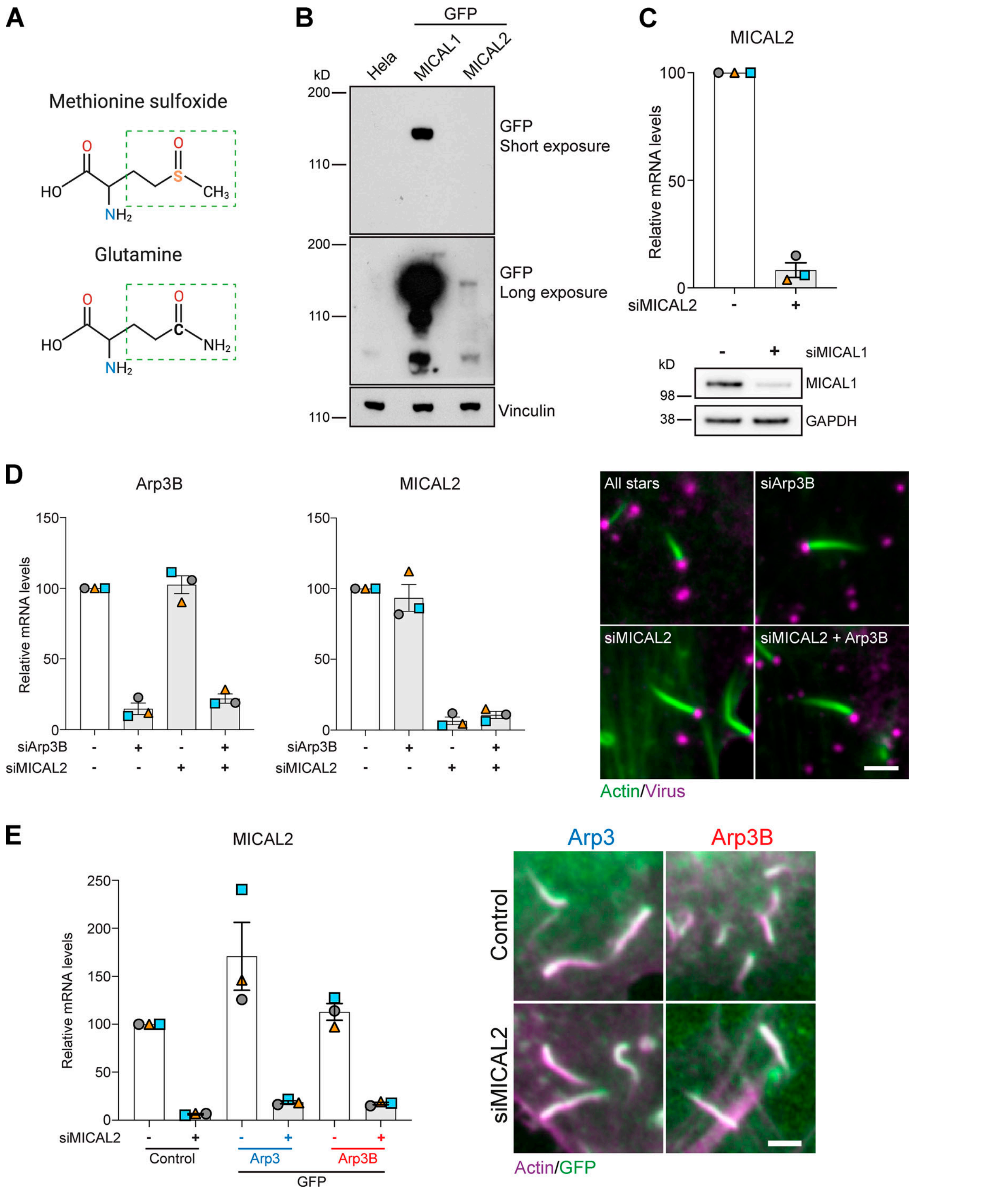


Figure S3. **MICAL2 regulates actin tail length. (A)** Condensed chemical formula of Met-SO and glutamine. The green rectangle highlights the side chain of both amino acids. **(B)** Immunoblot analysis of the level of stable GFP-tagged MICAL1 and MICAL2 expression in HeLa cells. **(C)** Immunoblot and RT-PCR analysis of the level of MICAL1 protein or MICAL2 mRNA, respectively, in HeLa cells treated with MICAL1 and MICAL2 siRNA. **(D)** RT-PCR analysis of the level of Arp3B and MICAL2 mRNA in cells treated with Arp3B and MICAL2 siRNA. The images show representative actin tails labeled with Alexa Fluor 488 phalloidin (green) induced by vaccinia (magenta) in cells treated with the indicated siRNA. **(E)** RT-PCR analysis of the level of MICAL2 mRNA in cells treated with MICAL2 siRNA and stably expressing GFP-tagged Arp3 (blue) or Arp3B (red). The images show representative actin tails labeled with Alexa Fluor 568 phalloidin (magenta) in cells stably expressing GFP-tagged Arp3 or Arp3B (green) and treated with MICAL2 siRNA. Error bars represent SEM from three independent experiments. Scale bars = 2.5 µm.

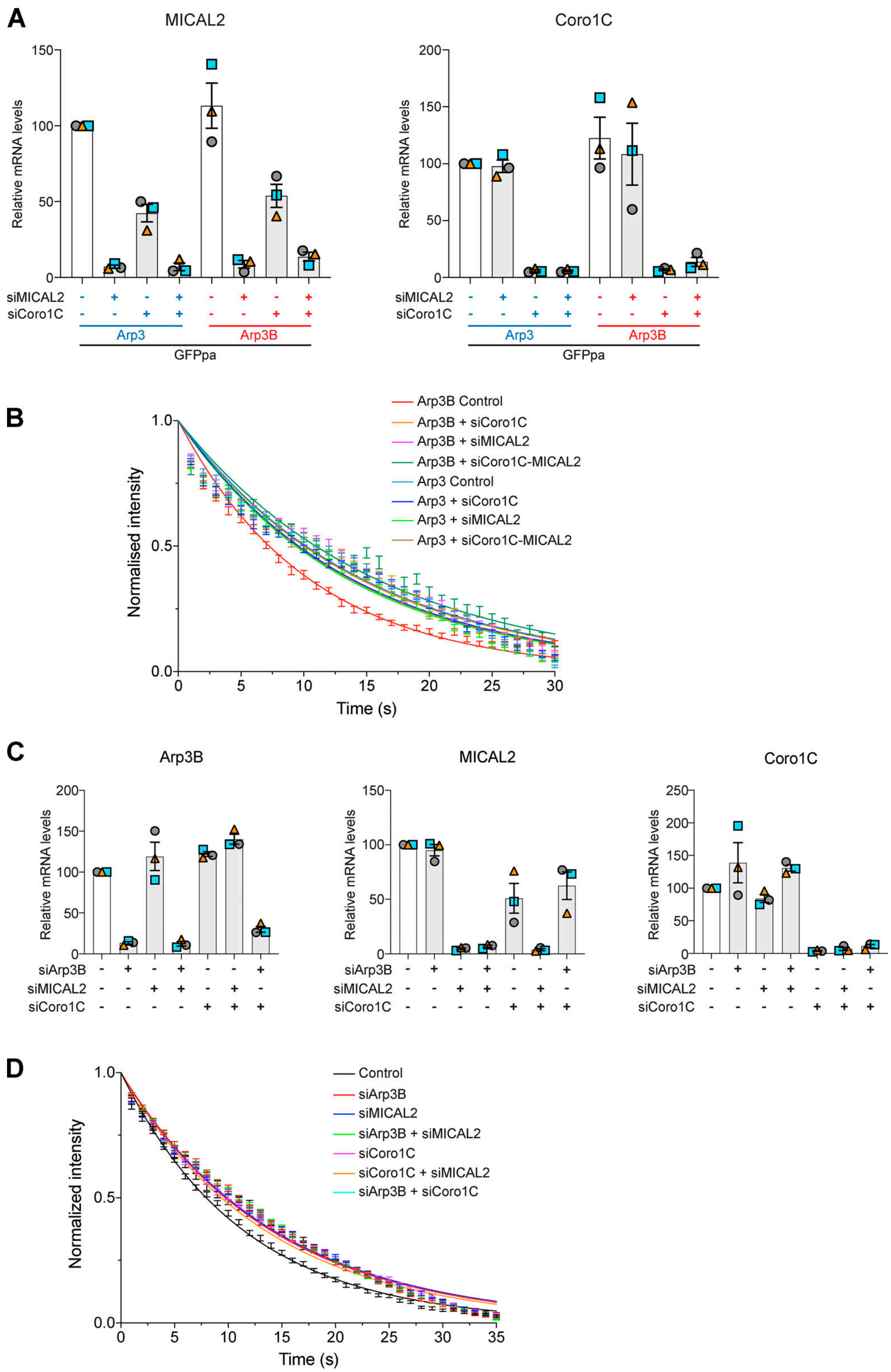


Figure S4. **Coronin 1C and MICAL2 cooperate to promote actin disassembly. (A)** RT-PCR analysis of the level of MICAL2 and coronin 1C mRNA in HeLa cells stably expressing GFP$^{PA}$-tagged Arp3 and Arp3B and treated with MICAL2 and/or coronin 1C (Coro1C) siRNA. **(B)** Quantification of the half-life of photoactivated GFP$^{PA}$-tagged Arp3 or Arp3B in actin tails in cells treated with coronin 1C and/or MICAL2 siRNA. **(C)** RT-PCR analysis of the level of MICAL2, Arp3B, and coronin 1C mRNA in HeLa cells stably expressing Cherry-GFP$^{PA}$-β-actin and treated with different combinations of MICAL2, Arp3B, and coronin 1C siRNA. **(D)** Quantification of the half-life of photoactivated Cherry-GFP$^{PA}$-β-actin in actin tails in cells treated with different combinations of Arp3B, coronin 1C, and MICAL2 siRNA. The graphs in B and D represent the best-fitting curve for each condition (continuous line), together with the average normalized intensity of the GFP signal at every time point (error bars represent the SEM for the indicated number of tails). Error bars for RT-PCR analysis represent SEM from three independent experiments.

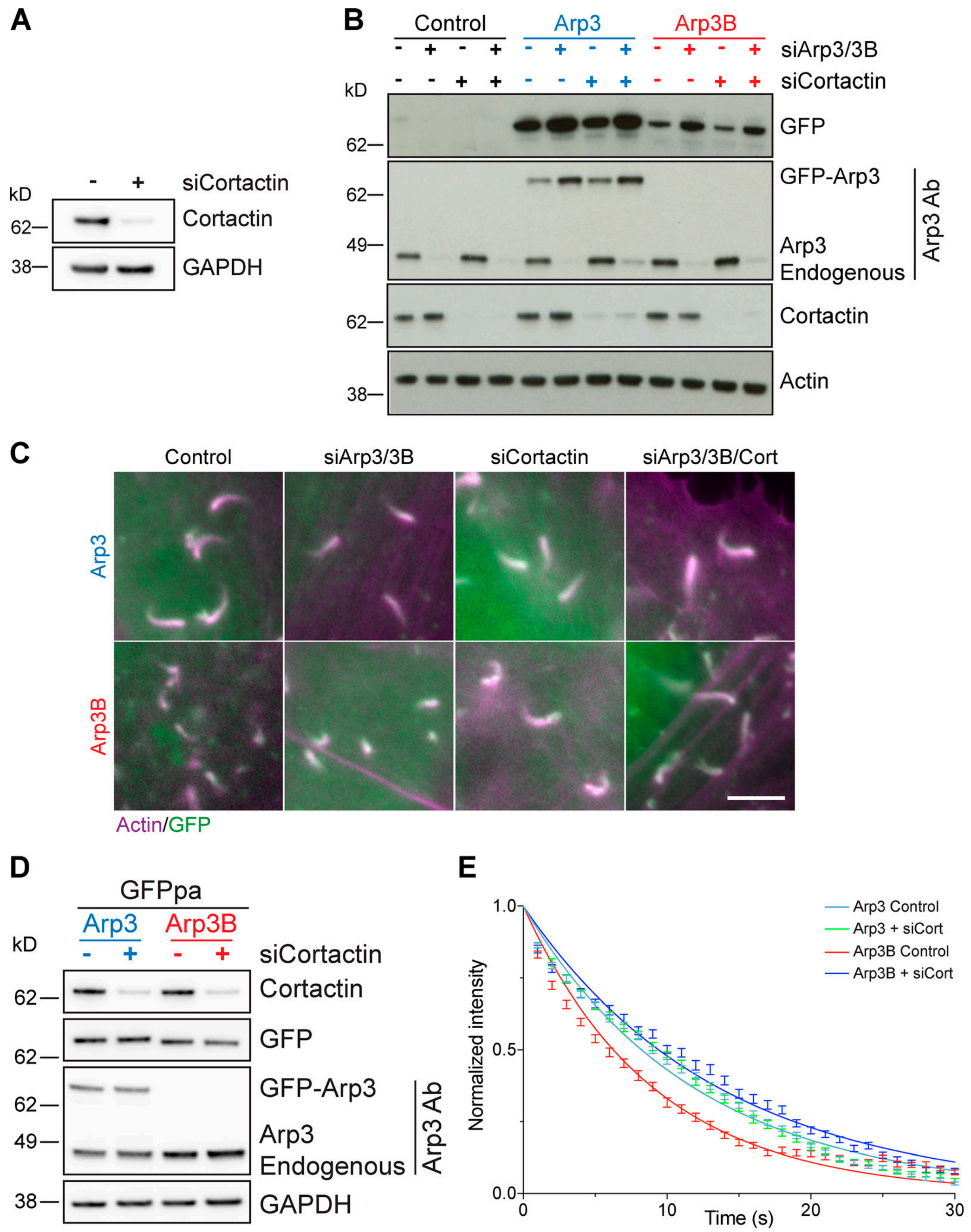


Figure S5. **Cortactin is required for enhanced turnover of Arp3B. (A)** The immunoblot shows the level of cortactin in HeLa cells treated with cortactin siRNA. **(B)** The immunoblot shows the expression level of RNAi-resistant GFP-tagged Arp3 and Arp3B in HeLa cells treated Arp3/Arp3B and cortactin siRNA. **(C)** Immunofluorescent images showing representative actin tails visualized with Alexa Fluor 568 phalloidin (magenta) in cells expressing GFP-tagged Arp3 and Arp3B (green) and treated with the indicated siRNA. Scale bar = 5 µm. **(D)** Immunoblot analysis of the level of cortactin in HeLa cells stably expressing $GFP^{PA}$-tagged Arp3 (blue) and Arp3B (red) and treated with cortactin siRNA. **(E)** Quantification of the half-life of photoactivated $GFP^{PA}$-tagged Arp3 or Arp3B in actin tails in cells treated with cortactin siRNA. The graph represents the best-fitting curve for each condition (continuous line), together with the average normalized intensity of the GFP signal at every time point (error bars represent the SEM for the indicated number of tails).

Video 1. **Time course of actin assembly (0.6 µM) in the presence of 2.5 nM Arp2/3 complex containing ARPC1A and ARPC5 and activated by 50 nM GST-(human WASp-VCA), acquired by TIRF microscopy.** The Arp3 isoform is indicated in each panel together with the time in seconds. The Arp3B data for the first 560 s is shown in Fig. 2 B. Images were taken every 5 s. Video plays at 17 frames per second. Scale bar = 15 µm.

Video 2. **Representative example of photoactivation of Cherry-GFP$^{PA}$-β-actin in a vaccinia actin tail in cells treated with control siRNA for 72 h (also see Fig. S2 C).** HeLa cells stably expressing Cherry-GFP$^{PA}$-β-actin (magenta before photoactivation and green after activation) were infected with vaccinia expressing YFP-A3 for 8 h. The time in seconds is indicated, and the scale bar = 15 µm. Images were taken every second. Video plays at 5 frames per second.

Video 3. **Representative example of photoactivation of Cherry-GFP$^{PA}$-β-actin in a vaccinia actin tail in cells treated with Arp3B siRNA for 72 h (also see Fig. S2 E).** HeLa cells stably expressing Cherry-GFP$^{PA}$-β-actin (magenta before photoactivation and green after activation) were infected with vaccinia expressing YFP-A3 for 8 h. The time in seconds is indicated, and the scale bar = 15 µm. Images were taken every second. Video plays at 5 frames per second.

Video 4. **Representative example of photoactivation of GFP$^{PA}$-Arp3 (appears green after activation) in a vaccinia actin tail, the analysis of which is presented in Fig. 2 E, with image stills in Fig. S2 E.** HeLa cells stably expressing GFP$^{PA}$-Arp3 were infected with vaccinia expressing YFP-A3 for 8 h before imaging. The time in seconds is indicated, and the scale bar = 15 µm. Images were taken every second. Video plays at 5 frames per second.

Video 5. **Representative example of photoactivation of GFP$^{PA}$-Arp3B (appears green after activation) in a vaccinia actin tail, the analysis of which is presented in Fig. 2 E, with image stills in Fig. S2 E.** HeLa cells stably expressing GFP$^{PA}$-Arp3B were infected with vaccinia expressing YFP-A3 for 8 h before imaging. The time in seconds is indicated, and the scale bar = 15 µm. Images were taken every second. Video plays at 5 frames per second.

Video 6. **Representative example of the recruitment of GFP-MICAL2 (green) to a vaccinia-induced actin tail visualized with LifeAct-iRFP670 (magenta) at 8 h after infection.** The time in seconds is indicated, and the scale bar = 5 µm. Images were taken every 3 s. Video plays at 5 frames per second.

Video 7. **Representative example of localization of GFP-MICAL1 (green) in a vaccinia-infected HeLa cell at 8 h after infection.** Virus-induced actin tails are visualized with LifeAct-iRFP670 (magenta). The time in seconds is indicated, and the scale bar = 5 µm. Images were taken every 3 s. Video plays at 5 frames per second.

Video 8. **Representative example of the recruitment of GFP-MICAL2 (green) to vaccinia-induced actin tails labeled with LifeAct-iRFP670 (magenta) at 8 h after infection in cells treated with control siRNA for 72 h.** The time in seconds is indicated, and the scale bar = 5 µm. Images were taken every 3 s. Video plays at 5 frames per second.

Video 9. **Lack of GFP-MICAL2 (green) recruitment to vaccinia-induced actin tails labeled with LifeAct-iRFP (magenta) in the absence of coronin 1C at 8 h after infection.** HeLa cells were treated with coronin 1C siRNA for 72 h before infection. The time in seconds is indicated, and the scale bar = 5 µm. Images were taken every 3 s. Video plays at 5 frames per second.

Video 10. **Representative example of the recruitment of GFP-coronin 1C and GFP-MICAL2 (green) to vaccinia-induced actin tails labeled with LifeAct-iRFP670 (magenta) at 8 h after infection in cells treated with cortactin siRNA for 72 h.** The time in seconds is indicated, and the scale bar = 5 µm. Images were taken every 3 s. Video plays at 5 frames per second.