## [Peer Review File · The Journal of Cell Biology]

MICAL2 enhances branched actin network disassembly by oxidizing Arp3B-containing Arp2/3 complexes

Chiara Galloni, Davide Carra, Jasmine Abella, Svend Kjær, Pavithra Singaravelu, David Barry, Naoko Kogata, Christophe Guérin, Laurent Blanchoin, and Michael Way

Corresponding Author(s): Michael Way, The Francis Crick Institute

Review Timeline:

Submission Date:	2021-02-06
Editorial Decision:	2021-04-06
Revision Received:	2021-04-27
Editorial Decision:	2021-05-11
Revision Received:	2021-05-18

Monitoring Editor: Bruce Goode

Scientific Editor: Lucia Morgado Palacin

Transaction Report:

DOI: <https://doi.org/10.1083/jcb.202102043>

April 6, 2021

Re: JCB manuscript #202102043

Dr. Michael Way
The Francis Crick Institute
Cellular signalling and cytoskeletal function Lab
The Francis Crick Institute
1 Midland Road
London NW1 1AT
United Kingdom

Dear Dr. Way,

Thank you for submitting your manuscript entitled "MICAL2 acts through Arp3B isoform-specific Arp2/3 complexes to destabilize branched actin networks". The manuscript was assessed by three expert reviewers, whose comments are appended to this letter. We invite you to submit a revision that addresses the reviewers' key concerns, as outlined here.

As you will see, the reviewers were overall very enthusiastic about your paper, but they raised a number of concerns that will need to be addressed before the paper would be deemed appropriate for publication in JCB. We hope that you will be able to address each of these concerns in full, including substantial new data to support the main conclusions of the study.

We hope, in particular, that you will be able to provide further insight into whether cortactin is needed for recruitment of MICAL2, as requested by rev #1, and into whether MICAL2 drives oxidation of Arp3B, as suggested by rev #3. However, we acknowledge that these experimental analyses are not strictly needed to support the main conclusions of the paper so we will not require such experiments for resubmission. Further discussion and/or comment/speculation would be fine. We hope that you will be able to address each of reviewers' other points, though.

GENERAL GUIDELINES:

Text limits: Character count for an Article is < 40,000, not including spaces. Count includes title page, abstract, introduction, results, discussion, acknowledgments, and figure legends. Count does not include materials and methods, references, tables, or supplemental legends.

Figures: Articles may have up to 10 main text figures. Figures must be prepared according to the policies outlined in our Instructions to Authors, under Data Presentation, <https://jcb.rupress.org/site/misc/ifora.xhtml>. All figures in accepted manuscripts will be screened prior to publication.

***IMPORTANT: It is JCB policy that if requested, original data images must be made available.

Failure to provide original images upon request will result in unavoidable delays in publication. Please ensure that you have access to all original microscopy and blot data images before submitting your revision.***

Supplemental information: There are strict limits on the allowable amount of supplemental data. Articles may have up to 5 supplemental figures. Up to 10 supplemental videos or flash animations are allowed. A summary of all supplemental material should appear at the end of the Materials and methods section.

As you may know, the typical timeframe for revisions is three to four months. However, we at JCB realize that the implementation of social distancing and shelter in place measures that limit spread of COVID-19 also pose challenges to scientific researchers. Lab closures especially are preventing scientists from conducting experiments to further their research. Therefore, JCB has waived the revision time limit. We recommend that you reach out to the editors once your lab has reopened to decide on an appropriate time frame for resubmission. Please note that papers are generally considered through only one revision cycle, so any revised manuscript will likely be either accepted or rejected.

Thank you for this interesting contribution to Journal of Cell Biology. You can contact us at the journal office with any questions, cellbio@rockefeller.edu.

Sincerely,

Bruce Goode
Monitoring Editor
Journal of Cell Biology

Lucia Morgado Palacin, PhD
Scientific Editor
Journal of Cell Biology

Reviewer #1 (Comments to the Authors (Required)):

This study investigates whether the two Arp3 paralogs (Arp3 and Arp3B) have different activities as a part of the Arp2/3 complex. As such, this work extends the previous accomplishments of this lab, where they characterized properties of other subunits of the Arp2/3 complex, Arpc1 and Arpc5, which also are encoded by two different genes. As a model system, the authors use the Arp2/3-dependent comet tail formation by vaccinia viruses in infected cells, and also conduct some biochemical experiments. They report that the branched actin networks that are assembled by Arp3B-containing Arp2/3 complexes undergo faster disassembly, because Arp3B can be oxidized at Met293 by MICAL2, whereas the Arp3-containing tails are more stable, as Arp3 cannot be oxidized.

In additional sets of experiments, authors show that both MICAL2 recruitment to tails and the enhanced actin disassembly depend on the MICAL2 interaction with coronin 1C. Furthermore, the authors show that cortactin also plays a role in this process, because Arp3B-containing tails are more stable in the absence of cortactin than in its presence. Overall, it is a very nice and important study, which is well supported by the data and clearly presented. These findings are expected to substantially advance our knowledge in this field.

My main concern is a lack of sufficient clarity about the roles of cortactin, especially, as compared with roles of coronin 1C. The analyses of cortactin's involvement are done not as thoroughly as for coronin 1C. For example, is MICAL2 recruited to tails in the absence of cortactin? If so, does cortactin interact with MICAL2? If not, how does it function? Despite a more shallow analysis of cortactin, as compared with coronin, the discussion is mostly focused on cortactin, but still does not achieve sufficient clarity. Thus, it has been established originally that cortactin stabilizes Arp2/3-dependent actin branches. To my knowledge, this activity of cortactin has not been disproved. In their previous work (Abella et al., 2016), this lab showed that (1) cortactin may or may not stabilize branches depending on a combination of the Arpc1 and Arpc5 paralogs, and (2) that cortactin can recruit coronins to facilitate debranching rather than to stabilize branches. In the current manuscript, the stabilizing role of cortactin is not a part of the picture, as if HeLa cells have only the ARPC1A/C5-containing Arp2/3 complexes, which according to the Abella et al. study are not stabilized by cortactin, but are instead destabilized by its ability to recruit coronin. Clearer discussion and, ideally, additional experiments to determine whether cortactin is needed for recruitment of MICAL2, would further improve this excellent study.

A few minor comments:

1. In figure S4D, the label "All stars" is puzzling until one digs deeply into Methods. I suggest relabeling the figure to show that this is a control siRNA.
2. In Fig. 2E and 7A, it would be helpful to state in the table whose half lives are shown, for example, as "Arp3/3B half-life", similar how it is done for actin ("Actin half-life").
3. Does Arp3 antibody recognize only the Arp3 paralog or both Arp3 and Arp3B? Without this information it is difficult to interpret gels.

Reviewer #2 (Comments to the Authors (Required)):

Summary:

Understanding how Arp2/3 complex-derived actin networks disassemble is an important and underexplored question. Here, Galloni, Carra et al. show that Arp3B-containing complexes are more likely to disassemble than Arp3-containing complexes. The authors also show that this is dependent on a single methionine residue in Arp3B, as well as on coronin-1C, cortactin, and the methionine monooxygenase MICAL2. The topic is of broad interest to cell biologists, and the finding that Arp2/3 subunit isoforms regulate network turnover is exciting. The experimental system used is appropriate (to be honest it's hard to imagine a system better suited to answering this question!). This paper will be a welcome contribution to the field, and outlined below are points related to the presentation of the data that should be addressed prior to publication:

--Major points:

1. Statistical analyses and data representation. While we do not doubt the conclusions of this work, we believe more rigorous statistical analyses and more transparency in the data would strengthen this work. This is especially critical for experiments with high variability, as seems the case here given that the average tail length in control cells varies by more than 50% between experiments (2 vs. 3.5 microns). The quantification of tail lengths uses an N of 300, where each comet tail counts individually. This is problematic for two reasons:

a. This may result in artificially deflated p values, as each comet tail is not an independent test of the hypothesis. We suggest the authors reconsider the statistical methodology, as an N of 3 (average each experiment and use experiment-level averages for statistics) may be more appropriate. Alternatively, more complex statistical approaches should be taken to incorporate the hierarchy of the data.

b. The bar graphs with standard error of the mean for an N of 300 will inevitably have small error bars, which is not helpful for understanding the breadth of the data. A scatter dot plot showing individual tail lengths would be more useful, as would showing experimental averages coordinated by color or shape. A simple webtool has been developed that makes this approach simple to execute (see Joachim Goedhart's recent MBoC paper: <https://www.molbiolcell.org/doi/10.1091/mbc.E20-09-0583>).

2. Red and Green images. Having only the merged images shown (without black and white panels of individual channels, as in Fig 1C, 1D, 3D, 4D, 5A, 6D and various supplemental data panels) can be a serious obstacle that will prevent color blind individuals from being able to assess the data. The magenta and green images in Figure 5B are much better and more accessible. We would love to see individual channels in gray and merged images in magenta and green in all cases, or at the very least, only magenta and green instead of red and green. Also, Fig. 1A does not include overlay of the channels; please provide.

3. Simplifying data interpretation for the reader. There are a number of places in the manuscript where lack of explanations and/or figure presentation make it hard for the reader to appreciate the logic of the experiments and/or the data. This should be remedied in the following ways:

a. A summary model would be very nice to help the reader organize all the data into a coherent story. Specifically, the model should include diagrams outlining the logic of the authors interpretation of the various knockdown results.

b. The data are consistent with the reasonable interpretation that the shorter tails are due to debranching, but this has not been explicitly shown. Clarifying that this is the interpretation rather than the data would make the logic of the paper easier to follow.

c. In figure 2, it would be helpful if Arp3/3B were always blue or red; they switch in 2A and 2C. Additionally, color coding some graphs (e.g. 3D, 4E) would help the reader. Similarly, coloring the key residue in Fig 4B based on where the residue came from would make the diagram much more useful to the reader.

d. It took a long time to puzzle out that the Arp3 antibody did not recognize Arp3B. Although this seems obvious in hindsight, please explain this up front to save time for other readers.

e. Figure 7A, 7B, and 7F: the plots are very difficult to interpret. Perhaps it would be better to put the plots in the supplement and instead graph the half-life information that is provided in the tables and use these as the display items in Fig. 7.

f. An (obvious) alternative explanation for the main findings is that the results are due to differential expression of various versions. This is not compatible with the data, but because it is such an obvious alternative hypothesis, it should be explained to the reader.

4. There seems to be a lot of variability in control tail lengths across experiments (Comparing Fig 5A and Fig 1), which makes recycling the same control in Fig 1 a bit concerning-- were all of those samples run in parallel? If so, this should be stated explicitly in the figure legend. If the experiments were not done in parallel, the experiment should be repeated in triplicate with the appropriate controls done in parallel.

5. There are a few places where it is unclear whether the experiments have been conducted multiple times. This should either be clarified or remedied:

a. Figure 2A shows a pyrene assay that appears to have only been done once. It seems appropriate, given the importance of the point the data is making, that this experiment should be replicated.

b. How many experiments were done for the data in Fig 2C, and what do the dots and error bars represent?

--Minor points:

i. Molecular weight markers on blots. Please provide the approximate size of the bands in the blots. This will help with interpretation of the data.

ii. Fig. 1A: Arp3 seems to localize to stress fibers (but not Arp3B); this should be discussed as it is not expected.

iii. It would be helpful to direct the reader to Fig S1 in the Fig 1B legend, because it is difficult to determine if siArp3B was successful from the given blot.

iv. Additional information on figure legends would be helpful, including the following:

It is unclear whether the definition of "control" varies with the experiment or not. Either way, the term "control" should be defined explicitly in the figure legends (e.g. does it refer to "wildtype" HeLa cells, or to HeLa cells expressing playing GFP, etc.)

e.g. what staining was used for actin in Fig 1 A and D?

What are the lines in figure 2D-E?

v. In figure 2B, insets/magnification would help.

vi. Why does it look like there was no GFP-MICAL2 in the input for Fig 6B? It was assumed that this was due to low expression levels, but should be addressed.

Reviewer #3 (Comments to the Authors (Required)):

This study explores the mechanism behind the discovery that Arp3 and Arp3B contribute differently to actin tail formation in vaccinia virus, yet seem to have a similar ability to stimulate actin nucleation in vitro. The authors use a combination of recombinant engineered Arp2/3 complex in

vitro assays, such as pyrene actin and TIRF and assays with vaccinia infected cells to determine the mechanism of how Arp3B promotes short actin tails. Overall, the study is clear and convincing and brings some novel biological insights into how Arp3B, a lesser expressed isoform of Arp3, is regulated by oxidation to affect actin branch turnover. The authors use Arp3- Arp3B chimeras and mutagenesis to discover that Arp3B has an essential methionine Met293, that when mutated to the corresponding residue in Arp3, converts the phenotype to a long tail with stable actin. They also show that the methionine monooxygenase, MICAL2, is recruited to actin branches by coronin-1c and is the likely enzyme responsible for the oxidation of Met293. They go on to implicate cortactin, as a scaffold for recruitment of coronin-1c, leading to rapid turnover of the actin tails. Overall, they provide convincing evidence that MICAL2 promotes oxidation of Arp3B and thus promotes rapid actin turnover in branched networks. This story provides a new mechanism for regulation of the stability of branched actin networks, via oxidation of Arp3B.

Points to consider:

- 1 While the proposed mechanisms are well supported by the data presented in this study, it is never directly shown that MICAL2 oxidises Arp3B. This may not be possible, due to the labile nature of oxidation- but the authors should at least comment on this.
2. What is the significance or purpose of having a very low-level expression of Arp3B, a protein that seems to promote dynamic actin turnover? Is there a cell or tissue-type that expresses higher Arp3B?
3. Bar graphs are not ideal for data display. Can the authors consider to plot the individual means of the 3 independent experiments on each of the bar graphs? This would at least allow determination of experimental variation across different independent runs.

1st Revision - Authors' Response to Reviewers: April 27, 2021

Response to reviewers' questions and comments are in blue italic text below.

Reviewer #1 (Comments to the Authors (Required)):

This study investigates whether the two Arp3 paralogs (Arp3 and Arp3B) have different activities as a part of the Arp2/3 complex. As such, this work extends the previous accomplishments of this lab, where they characterized properties of other subunits of the Arp2/3 complex, Arpc1 and Arpc5, which also are encoded by two different genes. As a model system, the authors use the Arp2/3-dependent comet tail formation by vaccinia viruses in infected cells, and also conduct some biochemical experiments. They report that the branched actin networks that are assembled by Arp3B-containing Arp2/3 complexes undergo faster disassembly, because Arp3B can be oxidized at Met293 by MICAL2, whereas the Arp3-containing tails are more stable, as Arp3 cannot be oxidized. In additional sets of experiments, authors show that both MICAL2 recruitment to tails and the enhanced actin disassembly depend on the MICAL2 interaction with coronin 1C. Furthermore, the authors show that cortactin also plays a role in this process, because Arp3B-containing tails are more stable in the absence of cortactin than in its presence. Overall, it is a very nice and important study, which is well supported by the data and clearly presented. These findings are expected to substantially advance our knowledge in this field.

My main concern is a lack of sufficient clarity about the roles of cortactin, especially, as compared with roles of coronin 1C. The analyses of cortactin's involvement are done not as thoroughly as for coronin 1C.

We agree that the molecular details of the role of cortactin are still not clear. Moreover, our analysis suggests that there is still much about coronin-1C that remains to be uncovered.

For example, is MICAL2 recruited to tails in the absence of cortactin?

We have added new data showing that loss of cortactin does not impact on the recruitment of MICAL2 or coronin-1C (Fig. 9A). This further strengthens the importance of cortactin in mediating the action of MICAL2 and coronin-1C on Arp3B containing complexes

If so, does cortactin interact with MICAL2? If not, how does it function?

We do not have an antibody against MICAL2 that works for western so are limited to using GFP-MICAL2 for GFP-trap pulldowns. While this works well to detect coronin-1C, we have been unable to detect cortactin in these pulldowns. This most likely points to a transient interaction that only occurs at actin branches that is impossible to capture in a pulldown that requires a stable interaction.

Despite a more shallow analysis of cortactin, as compared with coronin, the discussion is mostly focused on cortactin, but still does not achieve sufficient clarity. Thus, it has been established originally that cortactin stabilizes Arp2/3-dependent actin branches. To my knowledge, this activity of cortactin has not been disproved. In their previous work (Abella et al., 2016), this lab showed that (1) cortactin may or may not stabilize branches depending on a combination of the Arpc1 and Arpc5 paralogs, and (2) that cortactin can recruit coronins to facilitate debranching rather than to stabilize branches. In the current manuscript, the stabilizing role of cortactin is not a part of the picture, as if Hela cells have only the ARPC1A/C5-containing Arp2/3 complexes, which according to the Abella et al. study are not stabilized by cortactin, but are instead destabilized by its ability to recruit coronin. Clearer discussion and, ideally, additional experiments to determine whether cortactin is needed for recruitment of MICAL2, would further improve this excellent study.

In Abella et al., 2016 we showed that ARPC1A/ARPC5 containing complexes nucleated branches with lower stability, resulting in short actin tails. Conversely, ARPC1B/ARPC5L nucleated more stable branches producing longer actin tails. Since the presence of coronin was required for the short actin tail phenotype of ARPC1A/ARPC5 containing complexes (figure 5f NCB paper), we concluded that these complexes were likely more susceptible to coronin mediated debranching, explaining the lower stability of their nucleated branches. In contrast, as coronin depletion did not further increase the long tail phenotype induced by ARPC1B/ARPC5L containing complexes (figure 5d, f NCB paper), we therefore suggested that these complexes are intrinsically more resistant to coronin mediated debranching, explaining their higher stability.

The different tail length phenotype produced by ARPC1A/ARPC5 or ARPC1B/ARPC5L isoform specific complexes, as well as the different stability of their nucleated branches, were both dependent on the presence of cortactin. It is therefore clear that cortactin plays a role in determining the different susceptibility of isoform specific complexes towards Coronin-mediated debranching. In our proposed model in Abella et al., cortactin has weaker or stronger interactions with Arp2/3 complexes depending on the isoform and this then impacts on the ability of coronin to induce debranching. Overall, we still think that cortactin does stabilize Arp2/3 at branches, but the degree of stabilisation depends on the Arp2/3 isoform.

Our new data also show that cortactin is required for the faster debranching of Arp3B containing complexes, as well as their short tail phenotype, which in turn are dependent on the presence of coronin 1C and on Arp3B oxidation by MICAL2:Coronin1C. This further highlights the cooperation of cortactin and coronin in determining the different stability of isoform specific complexes. We have seen no evidence in any of our expts that Arp3B preferentially goes into particular isoform complexes. As Arp3B will distribute randomly into all Arp2/3 complexes, the ability of cortactin to preferentially stabilize ARPC1B/ARPC5L over ARPC1A/ARPC5 complexes is not an issue as the impact of MICAL2 on Arp3B will impact all

isoforms. Going forward we will investigate the hierarchical relationship between Arp3B and ARPC1/ARPC5 isoforms in controlling branched actin network dynamics. However, at this stage we think this is beyond the current study as it will require the generation of CRISPR edited cell lines expressing defined isoform combinations.

Fully understanding the relationship between cortactin, Coronins, MICAL2 and isoform specific complexes will take a lot more work including single molecule analysis in vitro and cryo-EM of actin branches with cortactin. We think such work is beyond the scope of the current study. Nevertheless, we have modified the relevant discussion section and also included a line in the abstract to bring out the importance of cortactin and raise the outstanding questions concerning its role.

A few minor comments:

1. In figure S4D, the label "All stars" is puzzling until one digs deeply into Methods. I suggest relabeling the figure to show that this is a control siRNA.

We have ensured we now only say control.

2. In Fig. 2E and 7A, it would be helpful to state in the table whose half lives are shown, for example, as "Arp3/3B half-life", similar how it is done for actin ("Actin half-life").

We have specified which half-lives are measured on the new scatter plots graphs, which now say "Arp3/ Arp3B half-life" as requested. We feel that adding this information in the table could be confusing as the first column of the table already indicates if the measured half-life is Arp3 or Arp3B (this is done for every siRNA treatment). Having the headline on the second column might confuse the reader on which half-life is being examined as it would conflict with what stated in the first column.

3. Does Arp3 antibody recognize only the Arp3 paralog or both Arp3 and Arp3B? Without this information it is difficult to interpret gels.

The antibody only sees Arp3. We have now indicated this in the methods and legend for Figure 1B.

Reviewer #2 (Comments to the Authors (Required)):

Summary: Understanding how Arp2/3 complex-derived actin networks disassemble is an important and underexplored question. Here, Galloni, Carra et al. show that Arp3B-containing complexes are more likely to disassemble than Arp3-containing complexes. The authors also show that this is dependent on a single methionine residue in Arp3B, as well as on coronin-1C,

cortactin, and the methionine monooxygenase MICAL2. The topic is of broad interest to cell biologists, and the finding that Arp2/3 subunit isoforms regulate network turnover is exciting. The experimental system used is appropriate (to be honest it's hard to imagine a system better suited to answering this question!). This paper will be a welcome contribution to the field, and outlined below are points related to the presentation of the data that should be addressed prior to publication:

--Major points:

1. Statistical analyses and data representation. While we do not doubt the conclusions of this work, we believe more rigorous statistical analyses and more transparency in the data would strengthen this work. This is especially critical for experiments with high variability, as seems the case here given that the average tail length in control cells varies by more than 50% between experiments (2 vs. 3.5 microns).

Examination of our previous papers shows that actin tail lengths do vary depending on the study. Some of this variation comes from the person measuring the tail, as deciding where the tail ends, depends on the exposure of the original image and whether it is fixed or live. We do try to standardise image collection but somehow everyone is different. For this reason, we make sure that any set of data for individual experiments is always collected by the same person. It is also more usual for all tail measurements in a paper to be collected by the first author. However, this project involved two PhD students (joint first authors) that did not overlap. As author contributions showed the difference in control tail lengths (2 vs. 3.5 microns) highlighted by the reviewer actually represents a difference between the two first authors. However, this is not a problem as all experiments (images and quantification) for individual expts were collected by the same person. Moreover, we always determine the significance of our phenotypes by comparing them with the appropriate internal control condition of each experiment, in order to account for potential tail length variability between different sets of experiments.

The quantification of tail lengths uses an N of 300, where each comet tail counts individually. This is problematic for two reasons:

a. This may result in artificially deflated p values, as each comet tail is not an independent test of the hypothesis. We suggest the authors reconsider the statistical methodology, as an N of 3 (average each experiment and use experiment-level averages for statistics) may be more appropriate. Alternatively, more complex statistical approaches should be taken to incorporate the hierarchy of the data.

We have now gone through all our data and provided new graphs with the correct statistics based on the true N value. Some P values have changed slightly but the overall conclusions

remain the same.

b. The bar graphs with standard error of the mean for an N of 300 will inevitably have small error bars, which is not helpful for understanding the breadth of the data. A scatter dot plot showing individual tail lengths would be more useful, as would showing experimental averages coordinated by color or shape. A simple webtool has been developed that makes this approach simple to execute (see Joachim Goedhart's recent MBoC paper: <https://www.molbiolcell.org/doi/10.1091/mbc.E20-09-0583>).

We were already planning to change the graphs during revision to show the spread of the data even before the reviewers' comment as it is clearly the way forward for better data representation. We have provided new graphs showing the spread of data with individual data sets in the same colour. Because we have 300 data points, it is not always possible to see the independent data sets because of overlap but as the means show there is no obvious bias in their distribution.

2. Red and Green images. Having only the merged images shown (without black and white panels of individual channels, as in Fig 1C, 1D, 3D, 4D, 5A, 6D and various supplemental data panels) can be a serious obstacle that will prevent color blind individuals from being able to assess the data. The magenta and green images in Figure 5B are much better and more accessible. We would love to see individual channels in gray and merged images in magenta and green in all cases, or at the very least, only magenta and green instead of red and green. Also, Fig. 1A does not include overlay of the channels; please provide.

I completely agree that black and white images are easier to see but space constraints (in main and supplemental figures) meant that we had to show merged images in many figures. We have now converted these red/green merged images to magenta/green. We have also added single black and white images where we had space. We were able to do this because we removed panels corresponding to over expression of GFP-tagged proteins when there was no RNAi knockdown. We did this as we also removed the same data from the graphs to help reduce their complexity as we felt the rescue data in knockdown cells is a better expt than over expression of GFP-tagged protein in a wild type background.

The data in Fig. 1A was originally duplicated in Fig. 1D, which also contained additional data for tail lengths in Arp3 RNAi treated cells expressing GFP-tagged Arp3 and Arp3B. To save space we have deleted the original Fig. 1A and replaced it with Fig. 1D and corresponding images.

3. Simplifying data interpretation for the reader. There are a number of places in the manuscript where lack of explanations and/or figure presentation make it hard for the reader to

appreciate the logic of the experiments and/or the data. This should be remedied in the following ways:

We agree that the paper is data heavy with many different conditions for some experiments. To help clarify things we have removed data representing overexpression of GFP-tagged ARP, Arp3B or their mutants in the absence of RNAi treatment in the more complex graphs (see point 2)

a. A summary model would be very nice to help the reader organize all the data into a coherent story. Specifically, the model should include diagrams outlining the logic of the authors interpretation of the various knockdown results.

We have provided a model at the end of the paper to bring everything together.

b. The data are consistent with the reasonable interpretation that the shorter tails are due to debranching, but this has not been explicitly shown. Clarifying that this is the interpretation rather than the data would make the logic of the paper easier to follow.

We have now clarified this issue in the text on page 8 - specifically “ Our observations demonstrate that coronin-1C is required to recruit MICAL2 to actin tails to promote the disassembly of Arp3B but not Arp3 actin branches. Enhanced Coronin1C-mediated debranching of oxidised Arp3B containing complexes would account for the faster turnover of Arp3B and the short actin tails generated by its overexpression”.

c. In figure 2, it would be helpful if Arp3/3B were always blue or red; they switch in 2A and 2C. Additionally, color coding some graphs (e.g. 3D, 4E) would help the reader. Similarly, coloring the key residue in Fig 4B based on where the residue came from would make the diagram much more useful to the reader.

Throughout the paper we have now put labels for Arp3 in blue and Arp3B in red to be consistent with the in vitro data in figure 2. We tried many variations to do the same with the data points but this made it hard to compare the means and data collected in individual experiments in the N=3. Nevertheless, the colours do help the graphs. We made the labels for the chimeras purple and as suggested by the reviewer have coloured the residues based on where they came from while keeping the Arp3 and Arp3B blue and red respectively.

d. It took a long time to puzzle out that the Arp3 antibody did not recognize Arp3B. Although this seems obvious in hindsight, please explain this up front to save time for other readers.

We have now indicated that the Arp3 antibody does not recognize Arp3B in the methods and legend for Figure 1B.

e. Figure 7A, 7B, and 7F: the plots are very difficult to interpret. Perhaps it would be better to put the plots in the supplement and instead graph the half-life information that is provided in the tables and use these as the display items in Fig. 7.

As requested, we have now provided half-life graphs showing the spread of data and tables in the main figures (Fig. 9 and 10) and moved the plots to supplemental figures 4 and 5. We have also provided the spread of data for all half-life plots in the relevant figures.

f. An (obvious) alternative explanation for the main findings is that the results are due to differential expression of various versions. This is not compatible with the data, but because it is such an obvious alternative hypothesis, it should be explained to the reader.

We use a mixed population rather than deriving clonal lines for the stable cell lines of our GFP tagged proteins. We have found that in many cases the level of GFP-tagged Arp3/3B “normalizes” to about the same amount as the endogenous protein. This is clearly evident in many of our immunoblots. Moreover, the levels of expression of the various versions of Arp3 and Arp3B do not correlate with their phenotypes, e.g. higher expression did not correlate with longer tails and vice versa.

4. There seems to be a lot of variability in control tail lengths across experiments (Comparing Fig 5A and Fig 1), which makes recycling the same control in Fig 1 a bit concerning-- were all of those samples run in parallel? If so, this should be stated explicitly in the figure legend. If the experiments were not done in parallel, the experiment should be repeated in triplicate with the appropriate controls done in parallel.

Regarding variation in actin tail length please see response to the reviewers first point. In revising the paper, we have replaced the original Fig. 1A with Fig. 1D to avoid data duplication. The data in Fig. 1 and Fig. 5 (now figure 6A in revised paper) were collected several years apart by the two co-first authors (see response to reviewers first point concerning tail length variability).

5. There are a few places where it is unclear whether the experiments have been conducted multiple times. This should either be clarified or remedied:

We have gone through all our data and ensured the number of replicates is indicated.

a. Figure 2A shows a pyrene assay that appears to have only been done once. It seems

appropriate, given the importance of the point the data is making, that this experiment should be replicated.

The pyrene experiments, which were collected in 2016 were performed twice at 3 different concentrations (12.5, 25 and 50 nM Arp2/3).

b. How many experiments were done for the data in Fig 2C, and what do the dots and error bars represent?

The TIRF experiments were performed twice and at 3 different concentrations (0.5, 2.5 and 5 nM Arp2/3). In Fig. 2C each data point represents the average of 14 and 15 actin “structures” for Arp3B-C1B-C5L and Arp3-C1B-C5L respectively, although in both cases, due to the nature of the assay, that number varies over time - there are fewer structures to analyse at the beginning of the experiment than at the end.

The pyrene and TIRF assays demonstrated that the Arp3 and Arp3B containing complexes have the same actin nucleation efficiency regardless of the concentration used. Moreover, this result holds true whether they are ARPC1B/C5L or ARPC1A/C5 complexes - which is effectively another independent assessment. Collectively these data show that Arp3B and Arp3 containing complexes have the same actin nucleation efficiency.

--Minor points:

i. Molecular weight markers on blots. Please provide the approximate size of the bands in the blots. This will help with interpretation of the data.

As requested, we have indicated the sizes of the molecular weight markers on the different panels.

ii. Fig. 1A: Arp3 seems to localize to stress fibers (but not Arp3B); this should be discussed as it is not expected.

I have no idea why Arp3 seems to co-localize with a stress fibre in Fig. 1A which was not apparent in any of the other Arp3 images. In revising the paper this image has now been removed so it doesn't seem to make sense to discuss it further given it is not seen in any of the other images.

iii. It would be helpful to direct the reader to Fig S1 in the Fig 1B legend, because it is difficult to determine if siArp3B was successful from the given blot.

As requested, we have directed the reader to Fig. S1A in the legend (now Fig. 1C in the revised text).

iv. Additional information on figure legends would be helpful, including the following:

It is unclear whether the definition of "control" varies with the experiment or not. Either way, the term "control" should be defined explicitly in the figure legends (e.g. does it refer to "wildtype" HeLa cells, or to HeLa cells expressing playing GFP, etc.)

We have now clarified the definition of control in the methods and in the opening paragraph of results on page 4.

e.g. what staining was used for actin in Fig 1 A and D?

We have indicated how actin is labelled in the figure legends

What are the lines in figure 2D-E?

We assume the reviewer means the line fits. For all relevant half-life graphs we have now indicated in the legend that “ the graphs represent the best fitting curve for each condition (continuous line) together with the average normalised intensity of the GFP signal at every timepoint (error bars represent the SEM for the indicated number of tails)”.

v. In figure 2B, insets/magnification would help.

We tried making enlarged inserts for the panel so help see what was going on. We tried several variations but none worked as well as removing first 60 sec panel, which did not containing many actin structures and enlarging the remaining panels.

vi. Why does it look like there was no GFP-MICAL2 in the input for Fig 6B? It was assumed that this was due to low expression levels, but should be addressed.

MICAL2 is expressed at very low levels in our stable cell line. This is evident in Fig. S3B which shows an immunoblot with GFP with a short and long exposure. In contrast MICAL1 is expressed well. If anything, this further illustrates just how isoform specific the coronin-1C interaction is. We have added a line in the results section on page 7 to clarify why MICAL2 is not seen in the input and direct the reader to Fig. S3B.

Reviewer #3 (Comments to the Authors (Required)):

This study explores the mechanism behind the discovery that Arp3 and Arp3B contribute differently to actin tail formation in vaccinia virus, yet seem to have a similar ability to

stimulate actin nucleation in vitro. The authors use a combination of recombinant engineered Arp2/3 complex in vitro assays, such as pyrene actin and TIRF and assays with vaccinia infected cells to determine the mechanism of how Arp3B promotes short actin tails. Overall, the study is clear and convincing and brings some novel biological insights into how Arp3B, a lesser expressed isoform of Arp3, is regulated by oxidation to affect actin branch turnover. The authors use Arp3- Arp3B chimeras and mutagenesis to discover that Arp3B has an essential methionine Met293, that when mutated to the corresponding residue in Arp3, converts the phenotype to a long tail with stable actin. They also show that the methionine monooxygenase, MICAL2, is recruited to actin branches by coronin-1c and is the likely enzyme responsible for the oxidation of Met293. They go on to implicate cortactin, as a scaffold for recruitment of coronin-1c, leading to rapid turnover of the actin tails. Overall, they provide convincing evidence that MICAL2 promotes oxidation of Arp3B and thus promotes rapid actin turnover in branched networks. This story provides a new mechanism for regulation of the stability of branched actin networks, via oxidation of Arp3B.

Points to consider:

1 While the proposed mechanisms are well supported by the data presented in this study, it is never directly shown that MICAL2 oxidises Arp3B. This may not be possible, due to the labile nature of oxidation- but the authors should at least comment on this.

Since we started the long road to trying to publish our study, we have been trying to formally demonstrate that MICAL2 oxidises Arp3B on Met293. Unfortunately, the relevant peptide containing Met293 is not detected in the mass spec even when using recombinant Arp2/3 and a variety of different enzymes for digestion. Because of this, we raised an antibody against a short peptide containing oxidised Met293 as this approach has worked in the past to detect oxidised actin (Grintsevich et al., 2016 NCB). Using this Arp3B Met293-OX antibody, we now provide immunoblots in Fig. 7 of GFP-trap pulldowns on our stable cell lines that demonstrate ARP3B is oxidised. This Arp3B Met293-OX antibody specifically recognised a band corresponding to GFP-Arp3B but not GFP-Arp3. Moreover, this signal is lost when Met293 is mutated to threonine the equivalent residue in Arp3. Knockdown of MICAL2 also reduces the signal. The signal does not vanish as it is likely that MICAL2 is significantly depleted but not completely eliminated.

2. What is the significance or purpose of having a very low-level expression of Arp3B, a protein that seems to promote dynamic actin turnover? Is there a cell or tissue-type that expresses higher Arp3B?

Examination of available proteomic data bases reveals that Arp3B expression is always lower, typically 5% or less than Arp3. We have no idea why this is the case but consistent with the

original northern blot data in Jay et al., 2000 and our own unpublished RNAscope data on mouse tissues, Arp3B is most highly expressed in brain (<https://pax-db.org>).

3. Bar graphs are not ideal for data display. Can the authors consider to plot the individual means of the 3 independent experiments on each of the bar graphs? This would at least allow determination of experimental variation across different independent runs.

We have replotted the graphs as we requested (also see previous comments to reviewer 2)

May 11, 2021

RE: JCB Manuscript #202102043R

Dr. Michael Way
The Francis Crick Institute
Cellular signalling and cytoskeletal function Lab
The Francis Crick Institute
1 Midland Road
London NW1 1AT
United Kingdom

Dear Dr. Way:

Thank you for submitting your revised manuscript entitled "MICAL2 acts through Arp3B isoform-specific Arp2/3 complexes to destabilize branched actin networks". As you will see, the reviewers are now supportive of publication. We would be happy to publish your paper in JCB pending final revisions necessary to meet our formatting guidelines (see details below).

To avoid unnecessary delays in the acceptance and publication of your paper, please read the following information carefully regarding final formatting and uploading of materials.

A. MANUSCRIPT FORMATTING:

Full guidelines are available on our Instructions for Authors page, <http://jcb.rupress.org/site/misc/ifora.xhtml>. Submission of a paper that exceeds these limits without prior discussion with the journal office will delay scheduling of your manuscript for publication.

- 1) Figure formatting: Scale bars must be present on all microscopy images, including inset magnifications.
- 2) Statistical analysis: Statistical methods should be explained in full in the materials and methods. For figures presenting pooled data the statistical measure should be defined in the figure legends. Please also be sure to indicate the statistical tests used in each of your experiments (both in the figure legend itself and in a separate methods section) as well as the parameters of the test (for example, if you ran a t-test, please indicate if it was one- or two-sided, etc.). Also, since you used parametric tests in your study (e.g. t-tests), you should have first determined whether the data was normally distributed before selecting that test. In the stats section of the methods, please indicate how you tested for normality. If you did not test for normality, you must state something to the effect that "Data distribution was assumed to be normal but this was not formally tested."
- 3) Abstract and title: Make the title concise but accessible to a general readership. While your current title will be appreciated by the specialists, we feel that a more concise title will be more accessible to a broader cell biology audience. Therefore, we suggest the following title: "MICAL2 enhances disassembly of branched actin networks through Arp3B"
- 4) Materials and methods: Should be comprehensive and not simply reference a previous

publication for details on how an experiment was performed. Please provide full descriptions (at least in brief) in the text for readers who may not have access to referenced manuscripts. The text should not refer to methods "...as previously described."

5) Please be sure to provide the sequences for all of your primers/oligos and RNAi constructs in the materials and methods. You must also indicate in the methods the source, species, and catalog numbers (where appropriate) for all of your antibodies.

6) Microscope image acquisition: The following information must be provided about the acquisition and processing of images:

a. Make and model of microscope

b. Type, magnification, and numerical aperture of the objective lenses

c. Temperature

d. imaging medium

e. Fluorochromes

f. Camera make and model

g. Acquisition software

h. Any software used for image processing subsequent to data acquisition. Please include details and types of operations involved (e.g., type of deconvolution, 3D reconstitutions, surface or volume rendering, gamma adjustments, etc.).

7) Supplemental materials: Please note that tables, like figures, should be provided as individual, editable files. A summary of all supplemental material should appear at the end of the Materials and methods section.

8) eTOC summary: A ~40-50 word summary that describes the context and significance of the findings for a general readership should be included on the title page. The statement should be written in the present tense and refer to the work in the third person. It should begin with "First author name(s) et al..." to match our preferred style.

9) A separate author contribution section is required following the Acknowledgments in all research manuscripts. All authors should be mentioned and designated by their first and middle initials and full surnames. We encourage use of the CRediT nomenclature (<https://casrai.org/credit/>).

10) ORCID IDs: ORCID IDs are unique identifiers allowing researchers to create a record of their various scholarly contributions in a single place. At resubmission of your final files, please consider providing an ORCID ID for as many contributing authors as possible.

FINAL FILES:

In order to accept and schedule your paper, we need you to upload the following materials to eJP. If you have any questions about the online submission of your final materials, please contact JCB's Supervising Manuscript Coordinator, Lindsey Hollander (lhollander@rockefeller.edu).

1) Electronic version of the text: An editable version of the final text is needed for copyediting (no PDFs).

2) High-resolution figure and video files: Individual high-resolution, editable figure files must be provided for each figure. Acceptable figure file formats are .eps, .ai, .psd, and .tif. JCB cannot accept PowerPoint files. All images must be at least 300 dpi for color, 600 dpi for greyscale and 1,200 dpi for

line art. Videos must be supplied as QuickTime files.

3) It is JCB policy that if requested, original data images must be made available to the editors. Please ensure that you have access to all original data images prior to final submission.

4) Cover images: If you have any striking images related to this story, we would be happy to consider them for inclusion on the cover or table of contents. Images should be uploaded as .tif or .eps files and must be at least 300 dpi resolution.

****The license to publish form must be signed by all authors before your manuscript can be sent to production (a link to the electronic license to publish form will be sent to each individual author).****

You can contact me or the scientific editor listed below at the journal office with any questions, jcellbiol@rockefeller.edu.

Thank you for this interesting contribution, I look forward to publishing your paper in The Journal of Cell Biology.

Sincerely,

Bruce Goode
Monitoring Editor
Journal of Cell Biology

Lucia Morgado Palacin, PhD
Scientific Editor
Journal of Cell Biology

Reviewer #1 (Comments to the Authors (Required)):

I am satisfied with the revision.

Reviewer #2 (Comments to the Authors (Required)):

Dr. Galloni, Dr. Carra, Dr. Way, and co-authors have done a beautiful job revising this manuscript. They have thoughtfully and thoroughly responded to each of the concerns previously raised. Congratulations on an interesting and exciting study!